# Benchmarking informatics workflows for data-independent acquisition single-cell proteomics

Jianwei Wang [1,3], Yi Huang [1,2,3], Fanghua Lu[1,2], Qinqin Xu[2], Zhuo Yang[2], Yirong Jiang[2], Shaowen Shi[1], Jianzhang Pan [1,2], Yi Yang [1,2] ✉ & Qun Fang [1,2] ✉

Recent years have seen a rise of single-cell proteomics by data-independent acquisition mass spectrometry (DIA MS). While diverse data analysis strategies have been reported in literature, their impact on the outcome of single-cell proteomic experiments has been rarely investigated. Here, we present a framework for benchmarking data analysis strategies for DIA-based single-cell proteomics. This framework provides a comprehensive comparison of popular DIA data analysis software tools and searching strategies, as well as a systematic evaluation of method combinations in subsequent informatic workflow, including sparsity reduction, missing value imputation, normalization, batch effect correction, and differential expression analysis. Benchmarking on simulated single-cell samples consisting of mixed proteomes and real single-cell samples with a spike-in scheme, recommendations are provided for the data analysis for DIA-based single-cell proteomics.

Single-cell proteomics allows for the precise revelation of the heterogeneity of proteomes between individual cells, which is neglected or masked in conventional bulk analysis[1–4]. Facing the challenge posed by the low abundance of proteins in single cells, efforts have been made in sample preparation, liquid chromatography (LC) separation, mass spectrometry (MS) acquisition, and data analysis, enabling the measurement of several thousand proteins in small subpopulations of cells and even in single mammalian cells[5–12]. Recently, the combination of trapped ion mobility spectrometry (TIMS) and data-independent acquisition (DIA) MS, namely diaPASEF, has been one of the most popular choices for single-cell proteomics[13–17]. Unlike data-dependent acquisition (DDA) approaches with stochastic precursor selection used in earlier single-cell proteomic studies, DIA facilitates data completeness by fragmenting the same sets of precursors in every sample. In addition, MS/MS acquisition of diaPASEF focuses on the most productive precursor population, excluding most singly charged contaminating ions. These features have been demonstrated to significantly improve the sensitivity of single-cell proteomic analysis[13].

DIA MS data are highly convoluted, and their interpretation relies on ingenious informatics solutions[18,19]. Typical DIA analysis methods require a spectral library that determines the space of peptides possibly detectable, as well as their retention time, ion mobility, and/or fragment patterns[20]. Spectral libraries can also be generated by deconvoluting the DIA data per se[21,22] or by in-silico prediction[23–26], enabling library-free DIA analysis. Different DIA data analysis solutions have been compared in a systematic way for bulk proteomics[27–31], and this has been extended to the single-cell level in a few recent studies[32,33]. It has been observed that the mass spectral data of single-cell samples have unique features, such as the loss of fragment ions and the blurred boundary between analyte signals and background[34]. These features are key factors in peptide identification and quantification. Thus, performance of routine DIA informatics solutions, including identification coverage and error rates, as well as quantitative precision and accuracy, requires specialized assessment at the single-cell level.

After protein identification and quantification, the subsequent data processing procedures are crucial for gaining meaningful insights

---

[1]Single-Cell Proteomics Research Center, and Zhejiang Key Laboratory of Intelligent Manufacturing for Functional Chemicals, ZJU-Hangzhou Global Scientific and Technological Innovation Center, Zhejiang University, Hangzhou, China. [2]Department of Chemistry, Zhejiang University, Hangzhou, China. [3]These authors contributed equally: Jianwei Wang, Yi Huang. ✉e-mail: y_yi@zju.edu.cn; fangqun@zju.edu.cn

into molecular mechanisms from differentially expressed proteins. Common challenges in analyzing single-cell proteomic data include handling the presence of missing values and resolving the batch effects[2]. In single-cell proteomics, missing values tend to be more prevalent as the abundance of proteins may be close to or below the limit of detection. Moreover, systematic differences across batches may lead to data biases mistaken for cell heterogeneity. There have been prior efforts for identifying optimal data processing methods for bulk proteomic analysis, including performance evaluation of missing value imputation methods[35], benchmarking normalization and statistical tests[29,36], and optimizing workflows for differential expression analysis[37]. Methods for batch effect correction were comprehensively compared for single-cell transcriptomics[38]. Nevertheless, there has not been a systematic evaluation covering combinations of popular methods in the data analysis workflow for single-cell proteomics.

Here, we present a framework for benchmarking data analysis strategies for DIA-based single-cell proteomics. This framework enables a comprehensive comparison of popular DIA data analysis software tools and searching strategies, as well as a systematic evaluation of method combinations in subsequent informatics workflow, including sparsity reduction, missing value imputation, normalization, batch effect correction, and differential expression analysis. Benchmarking was conducted on simulated single-cell samples consisting of mixed proteomes and real single-cell samples with a spike-in scheme. Based on the benchmarking results, recommendations are provided for choosing data analysis workflows for DIA-based single-cell proteomics.

## Results

### Benchmarking DIA-MS data analysis solutions using simulated single-cell samples

Hybrid proteome samples of organisms mixed in defined proportions have been used as benchmarking samples for performance evaluation of quantitative proteomic methods[27]. In this study, we constructed simulated single-cell-level proteome samples consisting of tryptic digests of human HeLa cells, yeast and *Escherichia coli* proteins with different composition ratios. A sample consisting of 50% human, 25% yeast, and 25% *E. coli* was used as reference (S3). In the other four samples (S1, S2, S4, and S5), the human proteins were of equivalent abundance to the reference, while the yeast and *E. coli* proteins had expected ratios to the reference from 0.4 to 1.6. The total protein abundance of the three organisms injected into the LC-MS/MS was 200 pg to mimic the low input when analyzing single-cell proteome samples. Each sample was analyzed by diaPASEF using a timsTOF Pro 2 mass spectrometer with six technical replicates (repeated injections). These samples with ground-truth relative quantities allowed us to evaluate the quantification performance of different data analysis solutions at the single-cell level.

We surveyed the current mainstream solutions for DIA data analysis (Supplementary Note 1) and select three software tools, i.e., DIA-NN[39], Spectronaut[22], and PEAKS Studio[40], for benchmarking in this study. DIA-NN and Spectronaut have been the most popular choices for single-cell proteomic studies. PEAKS has been emerging as a sensitive and streamlined platform for DIA data analysis[41]. All these software support library-free and library-based DIA data analysis strategies. For library-free analysis, DIA-NN and PEAKS build predicted spectral libraries from protein sequences by deep learning, while Spectronaut generates spectral libraries implicitly from the DIA data per se by the directDIA workflow. For library-based analysis, Spectronaut (with Pulsar engine) and PEAKS can generate spectral libraries from DDA data, while FragPipe can perform DDA data searching and build spectral libraries for DIA-NN[42]. In this study, we built sample-specific spectral libraries (DDALib) by multiple DDA injections of individual organisms (2 ng) performed on the sample LC-MS/MS system as the DIA experiments. Spectral libraries were also composed

from community resources (PublicLib) using timsTOF data of HeLa, yeast, and *E. coli* digests (200 ng) with high-pH reversed-phase fractionation released by Sinitcyn et al.[43]. In addition, AlphaPeptDeep[26] was used as an external source of predicted spectral libraries at the whole-proteome scale of the organisms.

We first compared the performance of these searching strategies within each software (Supplementary Note 2, Supplementary Data 1, and Supplementary Figs. 1–6). For DIA-NN, the public spectral library-based strategy quantified more proteins and peptides (Supplementary Figs. 1a and 2a), and the library-free workflow yielded higher protein quantitative accuracy (Supplementary Figs. 1e and 2e). For Spectronaut, the sample-specific spectral library-based strategy outperformed the others in terms of detection capabilities (Supplementary Figs. 3a and 4a), and directDIA had an advantage in quantitative accuracy (Supplementary Figs. 3e and 4e). The public spectral library-based strategy showed the worst reproducibility with a high level of missing values (Supplementary Figs. 3b and 4b). For PEAKS, the sample-specific spectral library-based strategy outperformed the others in proteome coverage (Supplementary Figs. 5a and 6a), while the four strategies resulted in similar quantitative performance (Supplementary Figs. 5e and 6e).

Considering the potential limitation of spectral library availability in practical applications, we focused on the inter-software performance comparison without the need of external spectral libraries. Spectronaut (directDIA) quantified $3066 \pm 68$ proteins (mean ± standard deviation, sic passim) and $12\,082 \pm 610$ peptides per run, which is the highest numbers among the three software tools, followed by PEAKS ($2753 \pm 47$) ranked second at the protein level and DIA-NN ($11\,348 \pm 730$) at the peptide level (Fig. 1b and Supplementary Fig. 7a). From the 30 DIA runs, 3524 proteins were detected totally by Spectronaut. Among them, 57% (2013) proteins were shared in all the runs. DIA-NN resulted in the more missing values at the protein level with 48% (1468/3061) proteins shared in all the runs. With more stringent criteria on data completeness, the quantified protein numbers by DIA-NN decreased, while the gap between the Spectronaut and PEAKS were closing (Fig. 1c). Considering proteins and peptides shared in at least 50% runs in each sample, the three software shared 61% (2225/3635) proteins and 48% (8002/16 729) peptides (Fig. 1d and Supplementary Fig. 7c). Spectronaut detected 11% more (3194/2879) and 23% more (3194/2607) proteins than PEAKS and DIA-NN, respectively.

In terms of the quantification performance, coefficient of variation (CV) values of protein quantities were calculated among replicate runs to evaluate the precision (Supplementary Figs. 1d, 3d, and 5d). The median coefficient of variation (CV) values of protein quantities were 16.5–18.4% by DIA-NN, slightly smaller than 22.2–24.0% of Spectronaut. PEAKS quantified proteins less precisely (27.5–30.0%). A similar trend was observed at the peptide level (Supplementary Figs. 2d, 4d, and 6d). Fold change (FC) values protein and peptide quantities of samples S1, S2, S4, and S5 to the reference S3 were calculated based on average of the replicate runs of each sample (Fig. 1e). Proteins and peptides shared among the four strategies were used to compare the quantitative accuracy. In each of the 36 pairwise comparisons among the three software (of three organisms, with four samples against the reference), the outperforming software was determined with experimental median $\log_2$ FC values closer to the theoretical values and significant differences (t-test p-value < 0.05 and Cohen's $|d| > 0.2$) of the $\log_2$ FC distribution. DIA-NN outperformed the other software in 8 comparisons at the protein level and 3 at the peptide level. Spectronaut came in a close second with 6 prevailing comparisons at the protein level and 2 at the peptide level.

Inter-software performance comparison with other searching strategies is summarized in Supplementary Data 1. In brief, with the sample-specific spectral libraries, Spectronaut and PEAKS outperformed DIA-NN in terms of protein detection capabilities, while DIA-NN yielded higher quantification accuracy. With the public

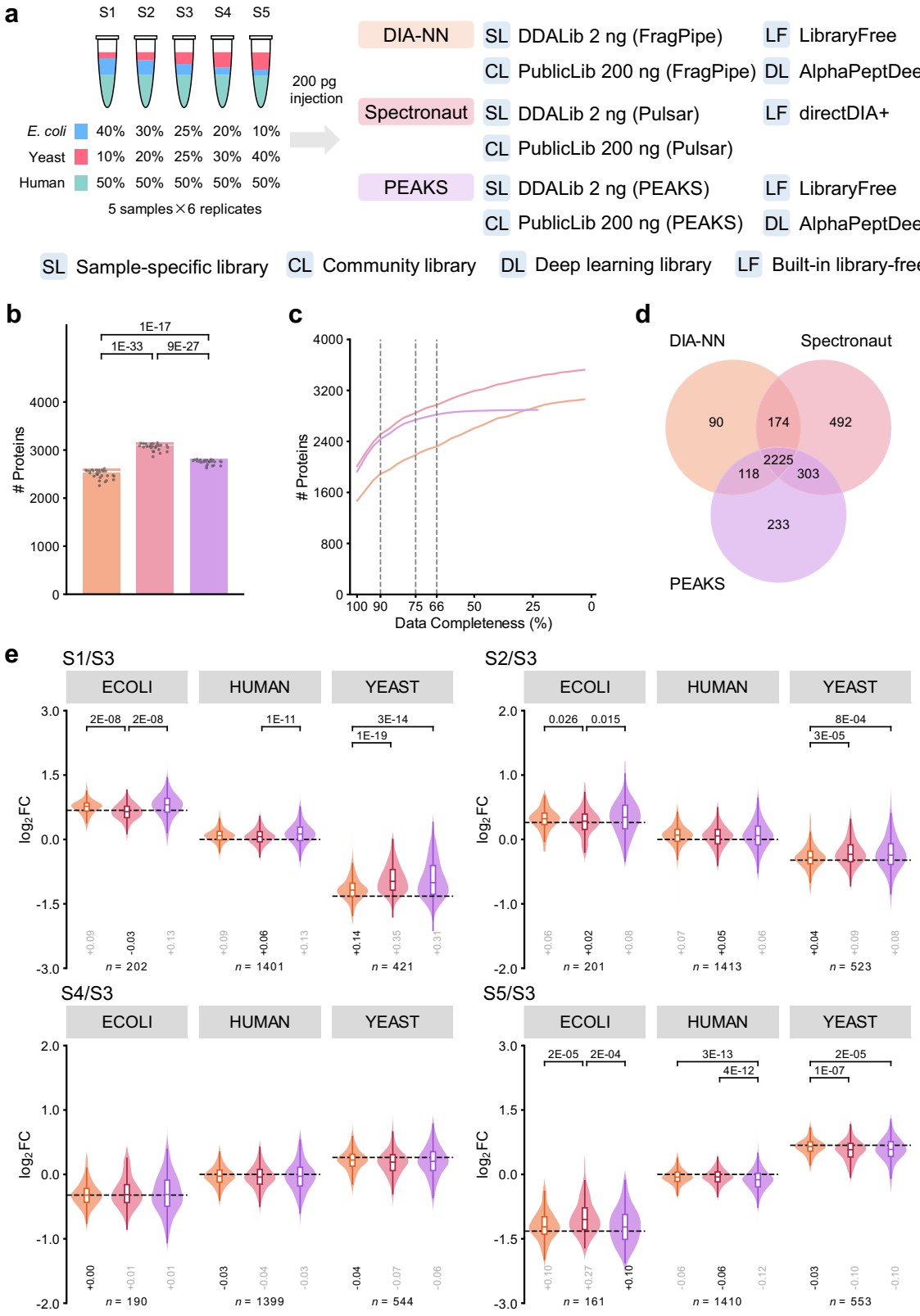

spectral libraries or predicted whole-proteome spectral libraries, DIA-NN still performed well for protein detection and quantification.

The above analyses were performed on the data of injection replicates of each sample. This design aims at evaluating the technical performance of each software with ground-truth samples, precluding bias originating from sample preparation. As for the latter, we generated a dataset of proteome samples prepared with independent

digestion at the single-cell level. Performance on this dataset using different software and searching strategies is summarized in Supplementary Data 2.

Modern DIA data analysis software allows transferring identifications from any one run to any of the others using the match-between-runs (MBR) algorithms[44], which enables peptides to be quantified across injections even without being initially detected in every single

**Fig. 1 | Performance comparison of different DIA data analysis software tools.**
**a** Construction of the benchmarking samples and the evaluated data analysis strategies. **b** Numbers of quantified proteins per run. The bars indicate the mean values and the error bars indicate the standard deviations. Significant differences (t-test p-value < 0.05, two-sided, no multiple comparison adjustments) are indicated. **c** Numbers of proteins quantified in at least specified percentages (data completeness) of runs. **d** Overlap of the proteins quantified in at least 50% runs. **e** Measured fold change (FC) values of protein quantities using sample S3 as reference. FC values were calculated only for proteins quantified in at least 3 runs for each sample of the comparison. Numbers (*n*) of proteins are indicated for each

species. The boxes mark the first and third quantile and the lines inside the boxes mark the median; the whiskers extend from the box to the farthest point lying within 1.5 times the inter-quartile range; outliers are not shown. The theoretical ratios are highlighted as dashed lines. Differences between the measured median FC values and theoretical values are indicated, among which the smallest ones are darkened. Significant differences (t-test p-value < 0.05 and Cohen's $|d| > 0.2$, two-sided, no multiple comparison adjustments) are indicated. In **b**–**e**, the data was analyzed using the library free strategy. Source data are provided as a Source Data file.

injection. The original aim of this feature is to mitigate the missing value problem. In single-cell proteomics, MBR across single-cell samples could lead to an increase in detected proteins[14]. Furthermore, researchers have co-searched single-cell data with those of higher-input samples, e.g., tens of cells, to enhance protein identification[45,46]. To assess the risk of erroneous transfer of identifications across runs, we performed a co-search test using organism-specific samples[47]. The mixed-organism samples S1–S5 were co-searched with three single-organism samples of human, yeast, and *E. coli*. The cross-organism searches could mimic the difficulties in analyzing largely heterogeneous samples, e.g., different cell types. Proteins and peptides originated from organisms absent in these samples were used to evaluate the level of potential false positive detection in the results (Supplementary Note 3). Compared with separate searches for each organism-specific sample group, co-searches with the mixed-organisms increased the identified proteins from the wrong organisms (Supplementary Figs. 8a and 9a). Although the wrong-organism hits were likely less intense in terms of median values, their quantity distributions shared a quite large overlap (Supplementary Data 1 and 2) and thus they could not be filtered with a quantity threshold. However, their high level of missing values indicated that false transfers were possibly spurious. Therefore, constraints of data completeness may rule out potential false positive hits partially (e.g., 75% data completeness could remove 10–83% entrapment hits despite 6–28% loss of the sample-specific proteins, Supplementary Data 1, Supplementary Figs. 8b and 9b). It was further observed that the sample-specific spectral libraries were beneficial in controlling false positive detection as they reduced the percentage of incorrect targets. However, since the generation of such spectral libraries requires some effort and may not always be possible, library-free strategies were adopted in subsequent benchmarking.

## Performance assessments of batch effect correction and differential expression analyses

Using the scheme described above, we performed another two DIA experiments of simulated single-cell-level samples with HeLa, yeast, and *E. coli* peptide standards. A combined dataset consisting of three batches of samples (batches of LC-MS analyses on different dates) were constructed for benchmarking batch effect correction and differential expression analyses. We started with the protein quantification result using the library-free mode of DIA-NN. Subsequent analysis steps included sparsity reduction, missing value imputation, normalization, batch effect correction, and, ultimately, statistical test for differential expression analyses (Fig. 2a). A set of popular algorithms were used for each step. For sparsity reduction, we applied: (1) no sparsity reduction (NoSR), (2) requiring >66% data completeness (SR66), (3) requiring >75% data completeness, and (4) requiring >90% data completeness per protein (SR90). For missing value imputation, we used: (1) replacing by 0 (Zero)[48], (2) replacing by half of the minimum in each row (HalfRowMin), (3) replacing by the mean in each row (RowMean), (4) replacing by the median in each row (RowMedian), (5) K-nearest neighbors (KNN)[49], (6) iterative low-rank singular value decomposition (IterativeSVD)[49], and (7) soft-thresholded singular value decomposition (SoftImpute)[50]. For normalization, we applied:

(1) unnormalized (this did not influence the potential internal normalization by DIA data analysis software with the recommended search settings), (2) median normalization, (3) sum normalization, (4) quantile normalization (QN), and (5) tail-robust quantile normalization (TRQN)[51]. For batch effect correction we used: (1) no batch correction (NoBC), (2) limma[52], (3) ComBat[53] using the parametric mode (ComBat-P), (4) ComBat[53] using the non-parametric mode (ComBat-NP), and (5) Scanorama[54]. Finally, statistical tests were used to probe for differentially abundant proteins: (1) Welch's t-test, (2) Wilcoxon–Mann–Whitney test (Wilcox), (3) limma[52] using the trend algorithm (limma-trend), (4) limma[52] using the voom algorithm (limma-voom), (5) edgeR[55] using the quasi-likelihood F-test (edgeR-QLF), (6) edgeR[55] using the likelihood ratio test (edgeR-LRT), and (7) DESeq2[56]. Detailed descriptions of these methods are present in Supplementary Note 4. By jointly assessing these methods for each step, a total of 4900 combinations were yielded.

We used several benchmarking metrics to assess the performance of batch effect correction and differential expression analyses. The adjusted Rand index (ARI)[38] was employed to evaluate the batch effect correction results, which computes similarity between discovered clusters (with the clustering parameters surveyed in Supplementary Note 5 and Supplementary Fig. 10) and ground-truth sample groups. After batch effect correction, the 7 statistical test methods were used to screen differential proteins between sample groups S4 and S2. For each method combination, the receiver operator characteristic (ROC) curve was obtained using the $-\log_{10}$ p-value output by the statistical test (adjusted by the Benjamini-Hochberg method) and the partial area under ROC curve (pAUC)[29] was determined. The pAUC value is a global performance indicator capturing the area under a low false positive rate (FPR) and can be interpreted as the average true positive rate (TPR) over the FPR range. In this study, we focus on the cases with FPR below 10% and thus the maximum possible pAUC value was 0.1. In a differential expression analysis task, differential proteins are typically detected at a given fold change (FC) and p-value threshold rather than an FPR range. Therefore, we further computed the accuracy, precision, recall, and F1-score with $|\log_2 FC| < \log_2 1.2$ (for S4/S2) and p-value < 0.05. Choices of these thresholds are explained in Supplementary Note 6 and Supplementary Fig. 11. Stricter p-value thresholds resulted in slightly fewer false positives but did not lead to significant changes (Supplementary Fig. 12). The method combinations were ranked based on ARI, pAUC, and F1-scores. The robustness of the metrics and ranking scheme was demonstrated by cross validation (Supplementary Note 7 and Supplementary Fig. 13).

We first investigated the influence of sparsity reduction. More strict sparsity reduction resulted in higher ARI, pAUC, and F1-scores (Supplementary Data 3). Despite the loss of quantified proteins after sparsity reduction, the higher data completeness facilitated the discrimination between truly differential proteins and false positives. The Pearson correlation coefficient (PCC) of the ranks of method combinations was computed across the sparsity reduction criteria. A median PCC of 0.86 was achieved (Supplementary Fig. 14a), indicating that ranking was stable regardless of sparsity reduction criteria.

In a comprehensive consideration of error rate control in protein identification (Supplementary Fig. 8) and detectability in differential

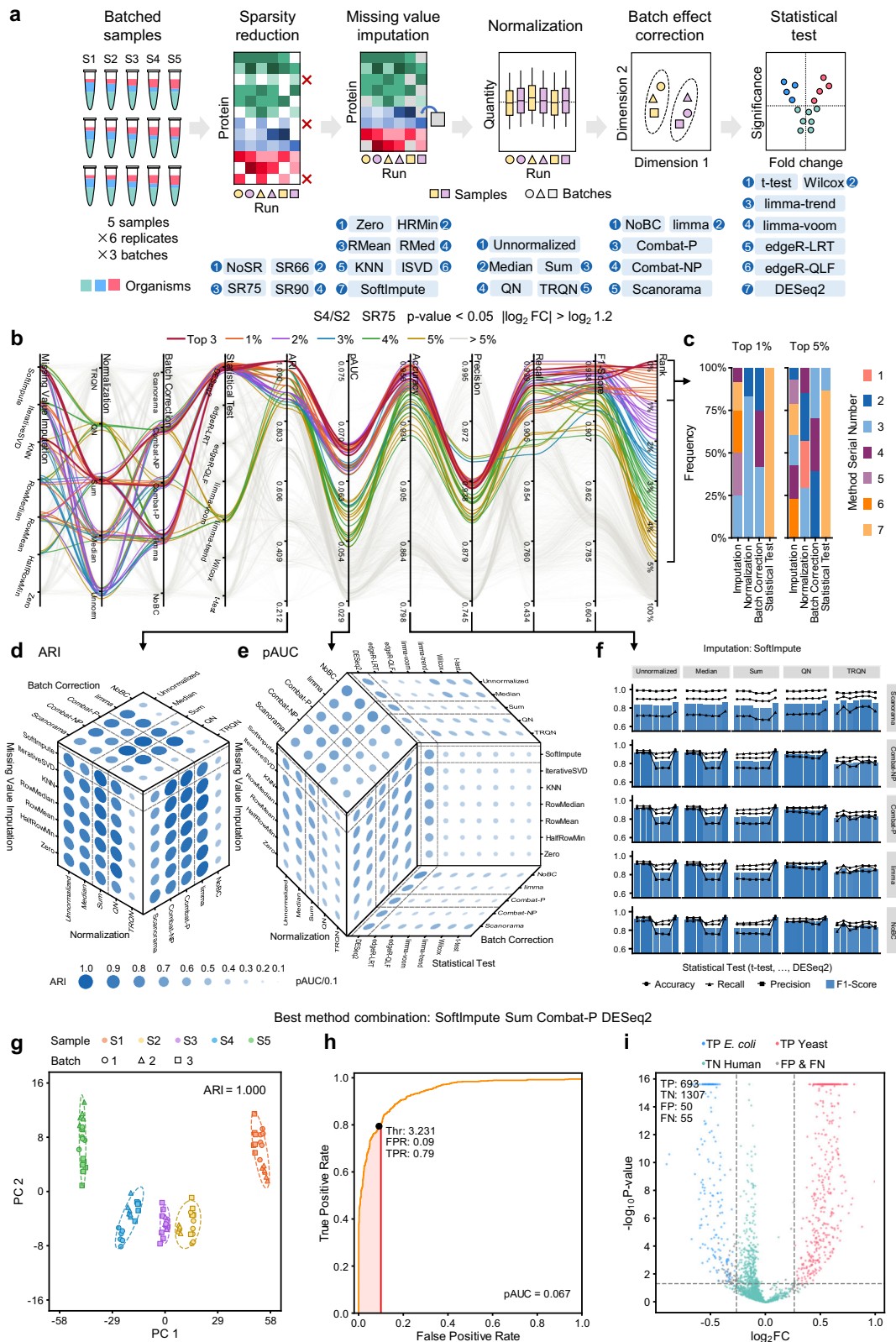

Best method combination: SoftImpute Sum Combat-P DESeq2

expression analysis, we focused on interpreting the results of SR75. The top 1% and 5 % method combinations, along with their metrics are visualized in Fig. 2b. The top 1% method combinations concentrated mostly in sum normalization, as well as Combat-P, Combat-NP, or limma for batch effect correction, and DESeq2 for statistical test (Fig. 2c). Among them, the best method combination (SoftImpute, Sum, Combat-P, DESeq2) resulted in an ARI of 1.0, a pAUC of 0.067,

and an F1-score of 93%. The variation of the metrics with different method choices are visualized in Figs. 2d–f. The ARI values were <0.5 without batch effect correction, while all the batch effect correction methods enhanced the clustering performance of the sample groups. The ARI values were more sensitive to normalization methods than missing value imputation. The pAUC values and F1-scores were mainly influenced by the choices of statistical tests, where DESeq2

**Fig. 2 | Performance comparison of different statistical method combinations for batch effect correction and differential expression analysis. a** Construction of the benchmarking dataset and the evaluated method combinations. **b** Parallel coordinate representation showing metrics using different method combinations. Line colors indicate the percentile rank of the method combinations. **c** Compositions of the top 1% and 5% method combinations in (**a**). Mappings of the serial numbers to detailed methods for each step are present in (**a**). **d** Adjusted Rand index (ARI) metrics. **e** Partial area under receiver operator characteristic curve (pAUC) metrics. In **c** and **d**, the metrics are visualized in a hyperbox, where each face displays the metrics with two steps variable and the other steps fixed to those of the best method combination. For the best method combination, the method choice in each step is marked with dashed lines. Dot sizes and colors indicate the metric values. **f** Accuracy (dots), recall (triangles), precision (squares), and F1-score (bars) metrics. Rows represent batch effect correction methods and columns represent normalization

methods. The other steps are those of the best method combination. **g** Clustering result of the five groups of samples visualized using principal component analysis for dimension reduction. The fill colors indicate the sample groups and the shape indicate the batches. The border colors indicate the clusters. **h** Receiver operator characteristic (ROC) curves using −log10 p-value as scores. The optimal cut-offs with false positive rate (FPR) ≤ 0.1 are marked using black dots with score threshold (Thr), FPR, and true positive rate (TPR) values indicated. **i** Volcano plots. Blue dots represent true positive (TP) *E. coli* proteins, red dots represent TP yeast proteins, green dots represent true negative (TN) human proteins, and gray dots represent false positive (FP) or false negative (FN) proteins. Benchmarks are performed on protein quantification results by DIA-NN. The data were processed starting with SR75. Differential analysis was performed between the S4 and S2 sample groups. Differential proteins are determined with p-value < 0.05 and $|\log_2 FC| > \log_2 1.2$.

fitted most of the imputation, normalization, and batch correction methods.

The same benchmarking workflow was performed on protein quantification results by the library-free mode of Spectronaut (Supplementary Fig. 15) and PEAKS (Supplementary Fig. 16). For Spectronaut, the top 1% method combinations concentrated mostly in median, sum, or no normalization, coupled with Combat-NP, Combat-P, or limma for batch effect correction, as well as DESeq2 and limma-trend for statistical test (Supplementary Fig. 15a, b). For PEAKS, limma was the optimal method for batch effect correction, and DESeq2 or limma-trend for statistical test (Supplementary Fig. 15a, b). With the same sparsity reduction criteria, a median PCC of 0.68 was obtained among the ranks of method combinations starting with quantification results by the three software (Supplementary Fig. 14c). Using the optimal method combinations, more truly differential proteins were screened from the DIA-NN result (Supplementary Fig. 14d), as a result of the higher accuracy in protein quantification. We further explored the performance of differential expression analysis between other sample groups (Supplementary Note 8).

It should be noted that biological covariates were provided to limma and Combat using batch effect correction in the above analyses. Although these covariates are given in the experimental settings of this study, we also simulated the scenario that covariates are not available in some single-cell studies (Supplementary Note 9 and Supplementary Data 4). We further tested the workflows with no imputation and ignoring the missing values for differential analyses as a strategy in some proteomics studies (Supplementary Note 10 and Supplementary Data 4).

## Mining the patterns of high-performing method combinations

To discover the patterns in the high-performing method combinations, we fitted the ranking results using a learning-to-rank model with extreme gradient boosting (XGBoost). Shapley additive explanations (SHAP) were performed on the top 25% method combinations. The resulting SHAP values quantify the contribution of each feature (method) to a prediction, and the SHAP interaction values extend this by measuring how pairs of features (methods of two steps) jointly influence predictions, capturing their combined effects beyond individual contributions. For the dataset of peptide standards, the statistical test step contributed the most to the ranking of method combinations (Supplementary Fig. 17a, b), where DESeq2 and limma-trend is the method choices with highest SHAP values (Supplementary Fig. 17c). Interactions between statistical test and normalization were also the highest (Supplementary Fig. 17d), where the combination of DESeq2 and Sum yielded the highest gains, followed by cross combinations of DESeq2/limma-trend and median/no normalization. The largest proportion of contribution by the statistical test step is reasonable since the samples were aliquots taken from of a large amount of peptide standards and the batch effect was dominated by

the LC-MS instrument status, which minimizes the biases during digestion.

With the benchmarking framework established, we then performed another experiment to mimic the real variations of single-cell proteome samples which are actually prepared from very little amount of proteins subjected to independent digestion (250 pg proteins in total as starting materials). Three batches of data were generated, resulting in a dataset containing two batches of samples preparation and another one containing two batches differing in LC-MS instruments (timsTOF Pro 2 and timsTOF Pro, Fig. 3a). Ranks and metrics of the method combinations on the dataset are present in Supplementary Data 5. Still, we focused on interpreting the results of differential analyses between sample groups S4 and S2 starting with SR75 from the quantification matrix of DIA-NN. For normalization, QN yield the most positive gain (Fig. 3d). While DESeq2 was the best choice considering the statistical method alone, limma-voom and edgeR-LRT were also high-performing methods if coupled with QN (Fig. 3e). For batch effect correction, Combat-NP, Combat-P, and limma resulted in positive gains per se and coupled with QN. The contributions of Scanorama were less stable. Missing value imputation by zero led to a negative impact.

The results by Spectronaut and PEAKS can be interpreted in a similar manner (Supplementary Figs. 18 and 19). Common patterns included: normalization was the most important step (Fig. 3b, c, as well as Supplementary Figs. 18a, b, 19a, b); statistical tests and batch effect correction had strong interactions with normalization (Fig. 3e, Supplementary Figs. 18d and 19d); imputation was relatively less important. The key difference of these datasets from that of peptide standards was the systematic variations introduced by the sample preparation process (e.g., differences in digestion efficiency). These variations require normalization and batch effect correction to correct biases before statistical analysis.

## Performance evaluation on real single-cell samples

We further evaluated the generalizability of the above data analysis method recommendations in characterizing real single-cell proteome samples (Fig. 4a). MCF-7 cells were treated with doxorubicin, a potent genotoxic agent, or DMSO as a control. Single-cell proteome samples were prepared using a pick-up single-cell proteomics analysis (PiSPA) platform[14] based on a microfluidic liquid handling robot. For each of the treatment and control cell group, half of the samples were spiked with 20 pg yeast and 40 pg *E. coli* digests (1Y2E), and the others with 40 pg yeast and 20 pg *E. coli* digests (2Y1E). The spike-in samples were analyzed by diaPASEF on a timsTOF Pro mass spectrometer in 3 batches, yielding a dataset containing a total of 60 samples. Different from the simulated samples consisting of tryptic digests, the spike-in single-cell proteome samples contained more complex matrices sourced from the cells. Although we tried to pick cells of close sizes, the abundance of human proteins may not be constant due to cell heterogeneity. Therefore, we selected a subset of

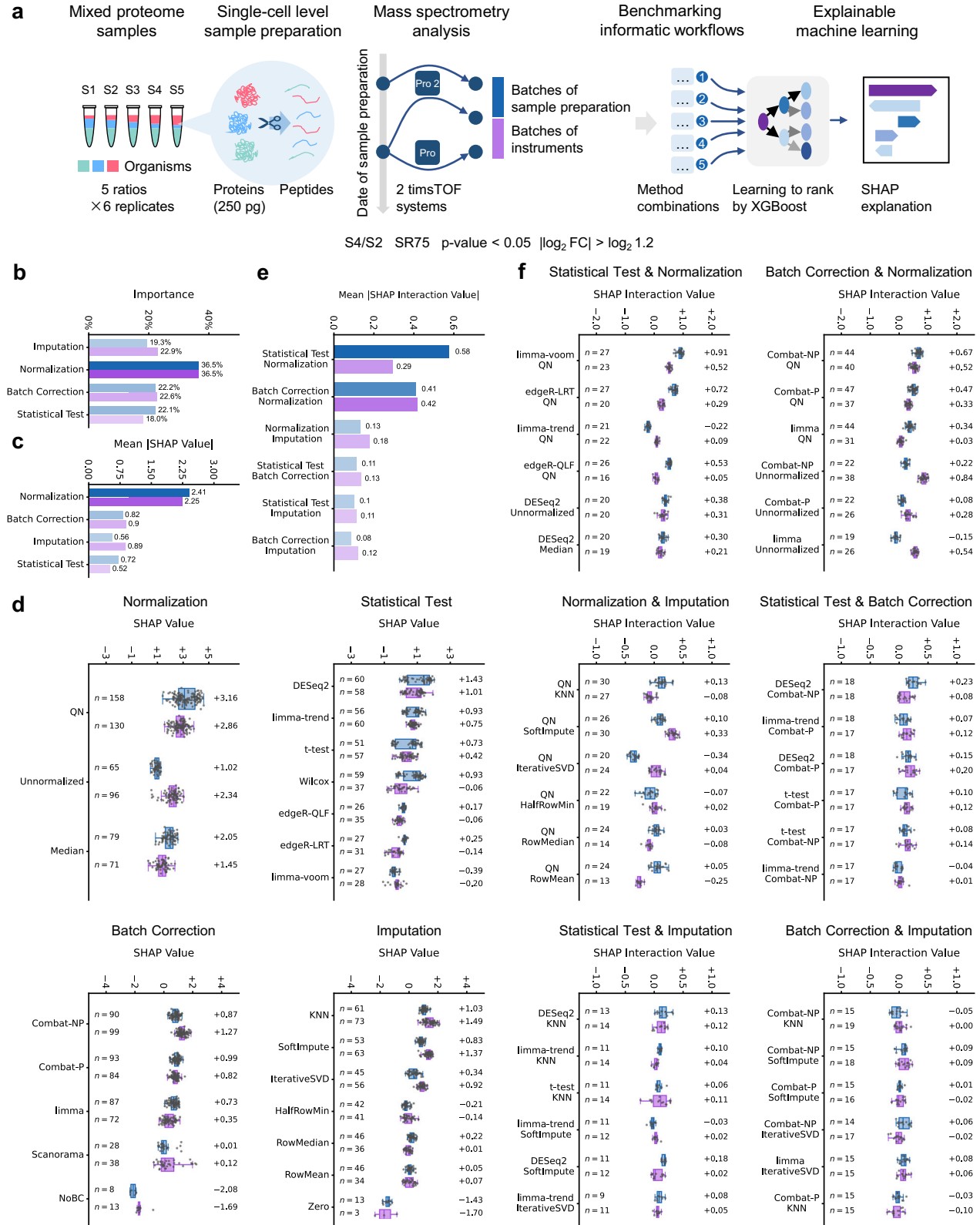

**Fig. 3 | Patterns of high-performing method combinations. a** Construction of the benchmarking dataset and the strategy for explaining the performance of the method combinations. **b** Feature importance of the model. **c** Mean absolute Shapley additive explanations (SHAP) values for each step. **d** SHAP values for the method choices in each step. **e** Mean absolute SHAP interaction values for each two steps. **f** SHAP interaction values for pairwise method choices in each two steps. In **d** and **f**, the boxes mark the first and third quantile and the lines inside the boxes mark the median; the whiskers extend from the box to the farthest point lying within 1.5 times the inter-quartile range. Individual data points are overlaid as dots. The median values and frequency (*n*) are indicated for each method choice. Benchmarks are performed on protein quantification results by DIA-NN. The data are processed starting with SR75. Differential analysis was performed between the S4 and S2 sample groups. Differential proteins are determined with p-value < 0.05 and |log$_2$ FC | > log$_2$ 1.2. Only the top 25% method combinations are subjected to SHAP explanation and the method choices with *n* < 3 are not shown. Source data are provided as a Source Data file.

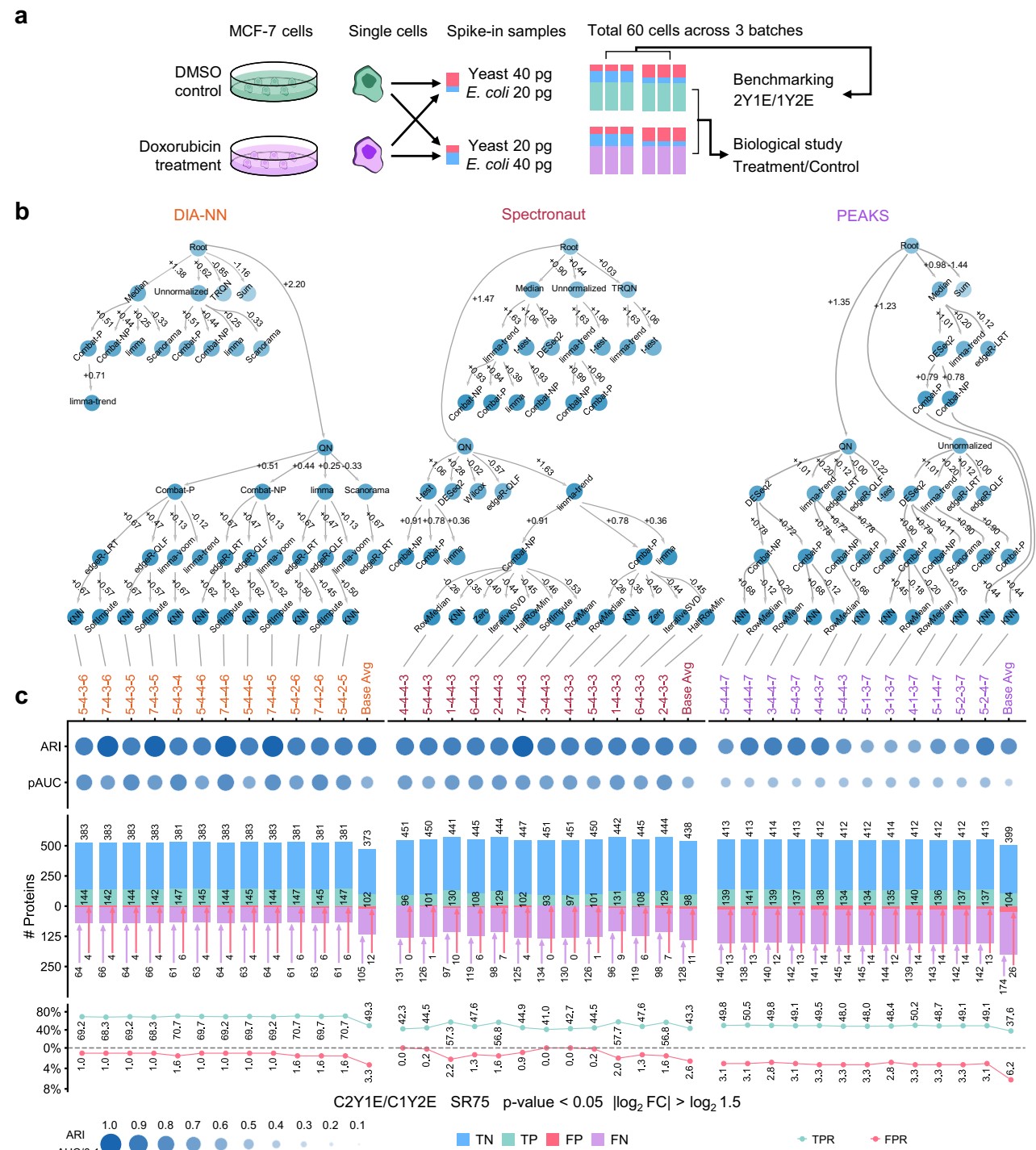

**Fig. 4 | Performance evaluation of high-performing method combinations on spike-in single-cell samples. a** Construction of the spike-in single-cell samples. **b** Selection of the high-performing method combinations based on the benchmarking results. **c** Performance of the selected high-performing method combinations on the spike-in single-cell samples. Metrics include: the adjusted Rand index (ARI) and partial area under curve (pAUC) values (indicated by dot sizes and colors), numbers of detected true negative (TN, blue bars), true positive (TP, green bars), false positive (FP, red bars), and false negative (FN, purple bars) proteins, as well as true positive rate (TPR, green lines) and false positive rate (FPR, red lines) values. Mappings of the serial numbers to detailed methods for each step are present in Fig. 2a. Average metrics of all the 1225 method combinations were used as a baseline (Base Avg). The data were processed starting from SR75. Differential analysis was performed between the C2Y1E and C1Y2E sample groups. Differential proteins are determined with p-value < 0.05 and |log₂ FC| > log₂ 1.5.

human proteins, including histones with cell size-independent copy numbers[57,58], as well as housekeeping proteins with probably stable expression[59,60] for the control groups (Supplementary Data 5). All the processing steps were applied on a matrix containing only the spike-ins and the subset of human proteins. In this setting, other proteins form the single cells should only provide the chemical background that challenges the overall analysis. The normalization and batch correction will work with the assumption that the real distributions of the samples were comparable, which they were not due to the variable cell sizes.

Based on the patterns of high-performing method combinations mined from the simulated samples, we established a strategy for the selection of method combinations (details in the Methods section). From the benchmarking result (S4/S2, SR75) on the quantification matrix by each software, 12 method combinations (~1% of all the method combinations) were selected by beam search (Fig. 4b). Performance of them was evaluated for screening differential proteins (p-value < 0.05 and |$\log_2$ FC| > $\log_2$ 1.5) between the two sample groups in the control subset (C2Y1E/C1Y2E). Average metrics of all the 1225 method combinations were used as a baseline, reflecting the expectation of random choices in the space of available method combinations. Generally, the selected method combinations resulted in higher TPR than the baseline for DIA-NN and PEAKS (Fig. 4c). Although the difference of TPR was small for Spectronaut, FPR was reduced for all three software. Results of other sparsity reduction criteria are presented in Supplementary Data 6, where false positives were at a low level (Supplementary Figs. 20). The results demonstrated the utility of the data analysis method selection strategy in real single-cell proteomic analyses.

## Similarities and differences of biological insights across data analysis workflows

The selected method combinations were then employed for screening differential proteins (p-value < 0.05 and |$\log_2$ FC| > $\log_2$ 1.5) between the treatment and control cells (T2Y1E/C2Y1E). For each method combination, Gene Ontology (GO)[61] and Reactome[62] pathway enrichment was performed on the differential proteins. Based on the enriched terms (adjusted p-value < 0.05), the method combinations were then clustered with the Jaccard distance (see Methods for details).

For the results with SR75, down-regulation of GO terms and Reactome pathways related to ribosomes, rRNA processing, translation, and cell-substrate junction were shared by the three software tools and most method combinations (Fig. 5, block 1). Some terms of nucleosome and nuclear transport were additionally enriched from DIA-NN results (block 2), while others of preribosome and nuclear speck were enriched from Spectronaut and PEAKS results (block 3). Oxidized DNA binding, as well as regulation of translation and some enzyme activities were detected uniquely by PEAKS (block 4). Up-regulation of GO terms related to mitochondrial matrix, as well as NAD and NADP metabolic process were discovered commonly (block 5). GO terms related to detoxification, oxidoreductase activity, some other small molecule catabolic process, as well as some organelles and granules were up-regulated in DIA-NN and/or PEAKS results (block 6). Spectronaut yielded up-regulated terms of amino acid transmembrane transport, nucleotide metabolic process, and microvillus components (block 7). Half of workflows starting with DIA-NN results detected the up-regulated of the TP53 pathway that regulates the transcription of metabolic genes (block 8).

The enriched results by workflows starting with different sparsity reduction conditions are presented in Supplementary Data 7 and Supplementary Fig. 21. From the protein quantification results by DIA-NN, although workflows starting with SR75 screened fewer differential proteins than NoSR, they yielded more enriched terms (at median). It also increased the stability of enriched terms across different workflows for all the software. The results stressed again the necessity to control data completeness, which could rule out potentially false hits.

Doxorubicin is known to intercalate with DNA base pairs, which eventually leads to DNA damage and the generation of reactive oxygen species (ROS)[63]. The p53 protein can be activated by DNA damage to induce a cell cycle arrest for damage repair[64,65], related to the enriched terms of DNA recombination, DNA packaging, and nucleosome. DNA damage can also induce a global decrease in translation and ribosome stalling through a p53-independent mechanism[66], consistent with the corresponding down-regulated terms. In addition, the enrichment results indicated that the cells upregulated energy production and

detoxification pathways to counteract oxidative stress and drug-induced toxicity. Amino acid transporters are essential for balancing intracellular amino acid pool for many cellular functions, including regulation of ROS levels and oxidative damage protection, which can enhance drug resistance[67]. This is supported by the up-regulated terms of amino acid transmembrane transporter activity. The alteration observed in microvillus, lysosomal, vacuolar and granule components, and other cellular components suggests the impacts of drug on cellular transport, metabolism, and secretion. Taking together the results by the data analysis workflows, the significant altered protein expression signatures can reflect basically the biological response in the presence of doxorubicin.

## Discussion

The recent technological and computational advances in DIA MS have powered label-free single-cell proteomics with promoted accessibility to the general proteomics audience and diverse core facilities[10,13,14,17]. While records have been chalked up in the numbers of identified proteins in single cells, diverse data analysis strategies have been employed in these reports, incorporating different software tools, spectral libraries, and downstream bioinformatics workflows. The impact of data analysis workflows on the outcome of single-cell proteomic experiments has been rarely investigated, which calls for dedicated benchmarking studies. Compared to prior efforts benchmarking DIA software and optimizing bioinformatics workflows for bulk proteomics[29,31,37], our study design provided a comprehensive comparison of DIA data analysis strategies for protein quantification at the single-cell level and embraced the complete statistical processing workflow in single-cell proteomics, including batch effect correction and differential expression analysis.

Our benchmarking included the two most popular software tools in DIA data analysis, i.e., DIA-NN and Spectronaut, as well as PEAKS Studio, a new choice that has not been used in published single-cell proteomics studies. Different types of spectral libraries and searching strategies were compared. Based on the overall performance of protein identification and quantification, we provided the following recommendations for single-cell DIA MS data analysis: (1) DIA-NN is preferred owing to its robustness across various spectral library types. Of note, DIA-NN is free for academic use and available to general proteomics researchers. (2) A project-specific spectral library built with low-input samples, if possible, is favorable, whereas a remote one of bulk samples is not advantageous. Spectronaut and PEAKS Studio could be alternative solutions when using a project-specific spectral library, although their results would be sensitive to the choice of libraries. (3) Co-searching with higher-input samples is risky for heterogeneous samples and close attention should be paid to the quality control. Data completeness is a useful criterion to control false identifications. (4) When a project-specific spectral library is not available (e.g., the samples are rare or resources for additional injections are not sufficient), Spectronaut directDIA is recommended as it yields more protein identifications, higher data completeness and comparable FPRs in differential protein analysis. The built-in prediction by DIA-NN is also an appropriate choice with high quantification accuracy and precision for library-free analysis. Currently, spectral library prediction using off-the-shelf models is not beneficial, as most of them are not optimized for single-cell samples. More precise prediction may be an option if the requirements and cost of training a model are affordable to general proteomics researchers in the future.

Our study conducted a systematic evaluation of statistical analysis workflows for single-cell proteomics. While clustering-based metrics such as ARI are commonly used to evaluate the performance of batch effect correction in single-cell omics in previous research[38], metrics based on the number of true/false differential proteins is more reliable as they reflect the performance of the workflow for differential expression analysis in practice. In this study, the method combinations

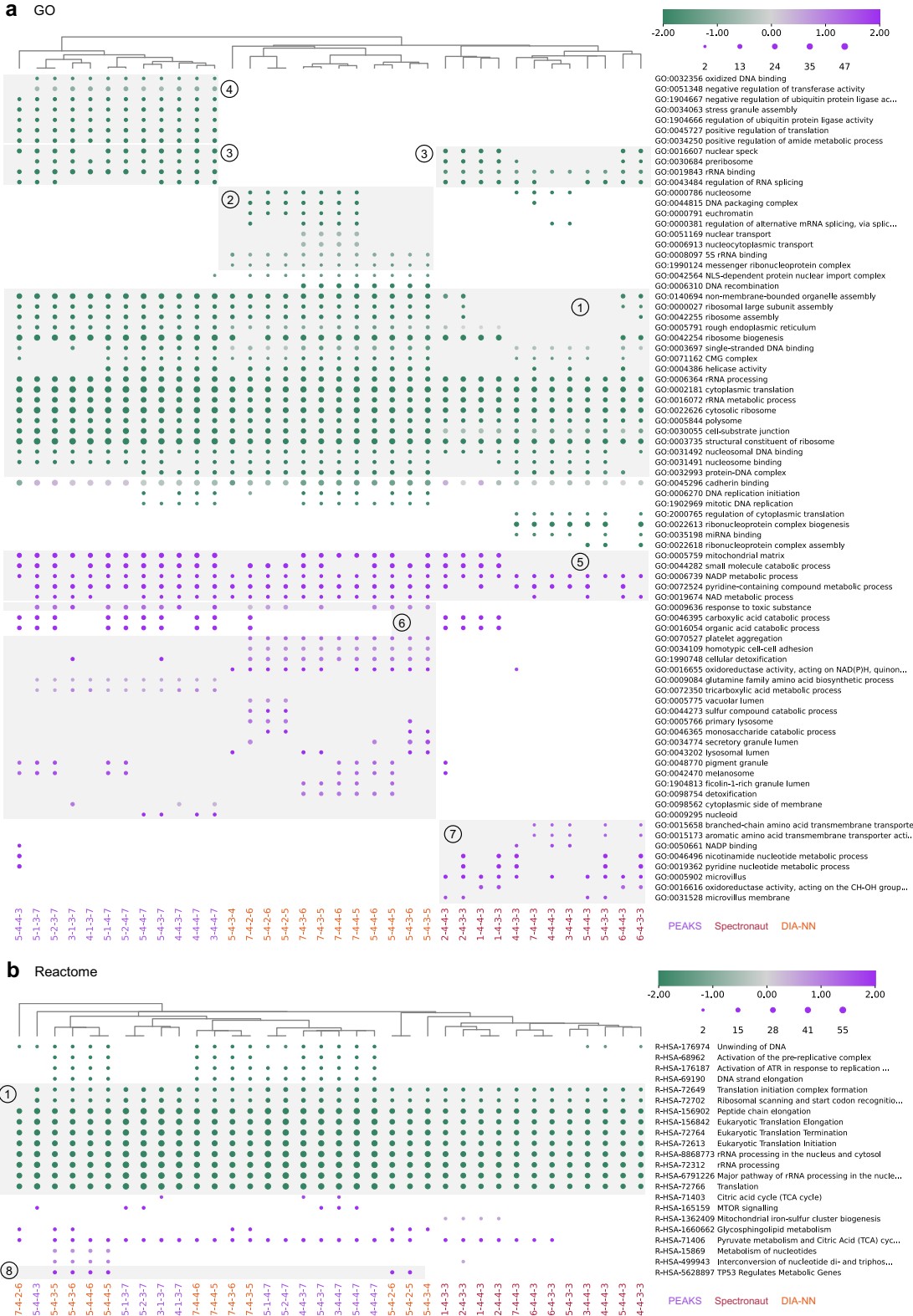

**Fig. 5 | Comparison of enrichment results from the differential proteins found by high-performing method combinations. a** Gene ontology (GO) enrichment results. **b** Reactome pathway enrichment results. Colors indicate Z-scores of terms. Dot sizes indicate the number of differential proteins. Columns were clustered with Jaccard distances. Mappings of the serial numbers to detailed methods for each step are present in Fig. 2a. The data were processed starting from SR75. Differential analysis was performed between the T2Y1E and C2Y1E sample groups. Differential proteins are determined with p-value < 0.05 and |log₂ FC| > log₂ 1.5.

were compared in a comprehensive consideration of ARI, pAUC, and F1-score, where the differential protein-based metrics served as a basis for comparison, supplemented with clustering-based metrics. Missing values in proteomics may originate from various mechanisms[35,48], including missing completely at random (MCAR) and missing at random (MAR) due to stochastic fluctuations (e.g., precursor selection in DDA and failed inter-run alignment in DIA), as well as abundance-dependent missing not at random (MNAR). In DIA single-cell datasets, analytes with a higher rate of missing values are probably close to the limit of detection or simply not present (false identification). After strict sparsity reduction, the remaining missing values could be better resolved by MAR/MCAR-devoted imputation methods. When analyzing the simulated single-cell samples, sum normalization was the optimal method since the total protein abundance was constant. However, this constraint is not met for the real single-cell samples, and choices of normalization methods depend on biological assumptions. As a standard part of many analysis pipelines for a wide range of data types[68], QN assumes the distribution of each sample is the same and can be used when no global changes may be of biological interest. For batch effect correction, ComBat (either parametric or not) and limma were preferred in most cases in our study, even if biological covariates were not available. It should be noted that the aim of batch effect correction for our single-cell proteomics workflow is to minimize the technical variability throughout the sample processing and MS acquisition (e.g., effects by analyzing the cells on different days) for differential expression analysis. This task is different from the scenarios of datasets integration (e.g., multiple experiments with different sequencing technologies or across labs)[38] and cell type identification. For statistical test, DESeq2 and limma-trend were present frequently in the high-performing method combinations. Nevertheless, the best statistical test methods were unstable across different DIA analysis software tools and sparsity reduction criteria.

Collectively, our study arrived at the following findings and recommendations for choosing data analysis workflows for DIA-based single-cell proteomics: (1) Sparsity reduction is a primacy in the whole workflow. For the homogeneous cell lines in this study, working with 75% data completeness is a good trade-off between gaining detected proteins and reducing the burden of missing value imputation. This will be more complicated when analyzing many different cell types, where missing values will be much higher due to cell-type-specific effects. (2) It is unfeasible to find a one-size-fits-all workflow that is optimal for various single-cell proteomics datasets. Researchers may use a set of high-performing method combinations demonstrated here (Fig. 4b) or determined from spike-in experiments more specific to their sample processing schema and instrument settings. (3) Data analysis workflows can result in similar or contrary biological insights. Parallel analysis by the set of high-performing method combinations could allow a comprehensive interpretation of single-cell proteomics results. A promising future development would be the integration of the results across workflows (e.g., using a voting model).

Notably, the results were produced by the latest versions of software that we had access at the time this study started. As the software is rapidly evolving, benchmarking of their performance needs an active update by the community beyond this study. We released the framework for benchmarking and analysis as an open-access tool named SCPDA. Although it was demonstrated here for timsTOF series instruments, the same processing scheme can be applied to optimizing data analysis workflows for Orbitrap or the recently emerging Astral mass spectrometers[12]. We expect that our work will provide resources aiding the data analysis of single-cell proteomics to the community.

## Methods

### Preparation of mixed-proteome samples

Two types of mixed-proteome samples were used for benchmarking.

(1) Mixed peptide standard samples. Pierce HeLa Protein Digest Standard (Thermo Fisher Scientific, 88329), MS Compatible Yeast Protein Extract Digest (Promega, V7461), and MassPREP *E. coli* Digest Standard (Waters, 186003196) were used to prepare the simulated single-cell-level proteome samples with different composition ratios (Supplementary Table 1). The total peptide concentration of the three organisms was 40 pg·μL$^{-1}$.

(2) Mixed-proteome samples subjected to independent digestion. MS Compatible Yeast Protein Extract Intact (Promega, V7341), as well as HeLa and *E. coli* protein extracts (obtained from Fudan University) were mixed with different composition ratios (Supplementary Table 1). For each sample, 250 pg proteins in total (12 nL) were transferred into an insert tube and processed like a real single-cell sample (see Preparation of single-cell samples below).

### Cell culture

Human breast adenocarcinoma MCF-7 cells (American Type Culture Collection, HTB-22) were authenticated by STR profiling and tested mycoplasma negative. Cells were cultured in DMEM medium (VivaCell, C311-0500) supplemented with 10% fetal bovine serum (VivaCell, C04001-500) and 1% penicillin/streptomycin solution (VivaCell, C3420-0100) and incubated at 37 °C in an atmosphere of 5% $CO_2$. Treatment with genotoxic agent doxorubicin (Sigma Aldrich, D1515) diluted in DMSO at final concentrations of 0.5 μM and with 0.1% (v/v) DMSO (Solarbio, D8371) as a control was performed for 24 h, respectively. For single-cell analysis, cells in a 6 cm dish were collected at 60–80% confluency using 0.25% trypsin with 0.02% EDTA (VivaCell, C3530-0500), and washed three times with phosphate buffer solution (PBS).

### Preparation of single-cell samples

Single-cell proteome samples were prepared using a PiSPA platform[14] based on a microfluidic liquid handling robot. Cell suspension (30–50 μL) was taken into a culture dish (35 mm diameter) and diluted by PBS to a final volume of 2 mL (corresponding to a cell suspension density of <10 000 mL$^{-1}$). Then the single target cell was picked up by the robot capillary probe into an insert tube.

After that, the robot was used to perform the subsequent sample pretreatment operations. First, 100 nL of 0.3% (w/v) RapiGest (Waters) was added to the insert tube with a sealing cap, reacted at 60 °C for 20 min for cell lysis, and cooled to room temperature; then 100 nL of 20 mM tris(2-carboxyethyl)phosphine was added to insert tube, reacted at room temperature for protein reduction; after that, 100 nL of 125 mM iodoacetamide was added, reacted at room temperature in the dark for 15 min; then 100 nL of mixed enzyme solution containing 0.05 μg·μL$^{-1}$ trypsin (Promega, V5111) and 0.05 μg·μL$^{-1}$ Lys-C (Promega, V1671) was added, reacted at 37 °C for 2 h; finally, 100 nL of 5% (v/v) formic acid was added to terminate the protein digestion for 30 min at room temperature.

Yeast and *E. coli* peptide standards with different composition ratios were spiked to the digested single-cell samples (Supplementary Table 2). The final single-cell sample was diluted by ultrapure water to a final volume of 5 μL and centrifuged at 500 × g for 3 min before LC-MS/MS injection.

### LC-MS/MS analysis

Two timsTOF systems were used for MS analysis.

(1) Most of the mixed-organism (all the samples of peptide standards and two batches of samples subjected to independent digestion) and single-organism samples were analyzed by a timsTOF Pro 2 mass spectrometer (Bruker) with an Vanquish NEO UHPLC system (Thermo Fisher Scientific). For each injection, 5 μL (200 pg) of peptides were loaded to an in-house capillary LC column (6 cm length, 50 μm inner diameter, 360 μm

outer diameter) packed with C18 particles (1.7 μm particle size, 120 Å pore size; Suzhou NanoMicro Technology) and separated with a 21 min gradient (Supplementary Table 3). Phase A was 0.1% formic acid in water, and phase B was 0.1% formic acid in 100% acetonitrile. The LC flow rate was 150 nL·min⁻¹ and the column temperature was 50 °C.

MS data were collected using Compass HyStar software (version 6.0). The DIA parallel accumulation serial fragmentation (diaPASEF) mode was used for MS and MS/MS acquisition. The range of the ion mobility $1/k_0$ was 0.75–1.3. The m/z acquisition range was 300–1500, the spray voltage was 1550 V, and the ion accumulation time was 166 ms. MS/MS was conducted with collision-induced dissociation and the collision energy was in the range of 20–59 eV, which varied with ion mobility. Six groups of 30 isolation windows with a width of 25 were set to cover a m/z acquisition range of 399–1149 (Supplementary Table 4).

(2) The rest of the mixed-organism samples (one batch of samples subjected to independent digestion) and the spike-in single-cell samples were analyzed by a timsTOF Pro mass spectrometer (Bruker) with an EASY-nLC 1200 system (Thermo Fisher Scientific). The sample was separated with a 21 min gradient (Supplementary Table 3). Phase A was 0.1% formic acid in water, and phase B was 0.1% formic acid in 80% acetonitrile. MS data were collected using Compass HyStar software (version 5.1). The spray voltage was 1750 V. Other conditions were the same as those for the mixed-organism samples.

## Spectral library building

Three types of spectral libraries were built for DIA data analysis.

(1) Sample-specific spectral libraries (DDALib). HeLa, yeast, and *E. coli* digest standards were individually analyzed by DDA experiments on the timsTOF Pro 2 system. For each injection, 1 μL (2 ng) of peptides was loaded. Other conditions were the same as those for the DIA experiments. For each sample, 30 replicates (repeated injections) were performed. These data were used to build the spectral libraries for the analysis of the mixed peptide standard samples. Similarly, the protein extracts digested in our lab were used to build the spectral libraries for the mixed-proteome samples subjected to independent digestion.

The raw DDA data were searched against Swiss-Prot/UniProtKB[69] databases of *Homo sapiens* (organism ID 9606, 20 422 entries, access date 2023-03-21), *Saccharomyces cerevisiae* strain ATCC 204508/S288c (organism ID 559292, 6727 entries, access date 2023-07-14), and *Escherichia coli* strain K12 (organism ID 83333, 4530 entries, access date 2023-07-03). For DIA-NN, FragPipe (version 22.0)[70] with MSFragger (version 4.1), IonQuant (version 1.10.27), and Philosopher (version 5.1.1) was used for protein identification and spectral library generation. For Spectronaut, spectral libraries were generated directly from the DDA data using the built-in function. For PEAKS, the DDA data were subjected to database searching and the spectral libraries were exported from the search results.

(2) Spectral libraries from community resources (PublicLib). Raw DDA data of HeLa, yeast, and *E. coli* digests (200 ng) acquired on timsTOF and released by Sinitcyn et al.[43] were searched using the same methods and parameters as those for building sample-specific spectral libraries.

(3) Predicted whole-proteome spectral libraries. AlphaPeptDeep (version 1.0.2)[26] was used for spectral library generation from the protein sequence databases. In each predicted spectrum, only the 12 highest peaks of at least 0.01 relative intensity were kept.

## DIA MS data analysis

Raw DIA data of the mixed-proteome samples were analyzed by DIA-NN, Spectronaut, and PEAKS. All of the software workflows were run using the default settings with modifications to make their results comparable. Trypsin/P was set as enzyme, and the maximum number of missed cleavages was set as 1. Carbamidomethylation (C) was specified as a fixed modification. Oxidation (M) and Acetylation (Protein N-term) were specified as variable modifications.

(1) DIA-NN (version 1.9.2)[39]. For the library-free analysis, deep learning-based in silico spectral library generation was enabled. For the library-based analysis, the sample-specific spectral library and the public spectral library built by FragPipe, as well as the predicted spectral library generated by AlphaPeptDeep, were used. Heuristic protein inference was enabled to make sure that no protein was present simultaneously in multiple protein groups. Detailed parameters are present in Supplementary Table 5.

The main report output by DIA-NN was processed the following filters: Q.Value (run-specific, at the precursor level) at 0.01 and Lib.Q.Value (for the respective library entry, at the precursor level) at 0.01, as well as PG.Q.Value at 0.05 and Lib.PG.Q.Value at 0.01 (same as above, at the protein group level). For the peptide-level quantification, the precursor-level results were aggregated by "Stripped.Sequence" and the quantity of each peptide was the sum of those of the top 3 corresponding precursors.

(2) Spectronaut (version 19.5.241126.62635, Biognosys)[22]. For the library-free analysis, the directDIA+ (Deep) workflow was used. For the library-based analysis, the sample-specific spectral library and the public spectral library built by Spectronaut were used. Detailed parameters are present in Supplementary Table 6. The exported report files were used for benchmarking. For the protein-level quantification, the columns "PG.ProteinAccessions" and "PG.Quantity" were used. For the peptide-level quantification, the columns "PEP.StrippedSequence" and "PEP.Quantity" were used.

(3) PEAKS Studio (version 12.0, Bioinformatics Solutions)[40]. For the library-free analysis, database searching was enabled and spectral library search was disabled. For the library-based analysis, database searching was disabled, while the sample-specific spectral library and the public spectral library built by PEAKS were used. De novo sequencing was disabled. Label-free quantification was switched to the high accuracy mode. In order to export the complete quantification results, protein significance filter was set to 0, protein fold change filter to 0.0–64.0, and used peptide filter to 0. Detailed parameters are present in Supplementary Table 7. The exported quantification report files (lfq.dia.proteins.csv and lfq.dia.peptides.csv) were used for benchmarking. For the protein-level quantification, the columns "Accession" and "Area [SampleName]" were used, where only the top protein in each protein group (indicated by the columns "Top" and "Protein Group") were kept. For the peptide-level quantification, the columns "Peptide" and "Area [SampleName]" were used. For the error rate estimation with single-organism samples, the exported identification report files (dia_db.proteins.csv and dia_db.peptides.csv for library-free analyses, or sl.proteins.csv and sl.peptides.csv for spectral library searches) were used.

## Data analysis workflows for differential expression analysis

The protein quantification matrices were subjected to data analysis workflows comprised of different combinations of sparsity reduction, missing value imputation, normalization, batch effect correction, and statistical test methods.

(1) Sparsity reduction. All proteins (NoSR) were kept; alternatively, only those present in at least 66% (SR66), 75% (SR75), or 90% (SR90) of all samples.

(2) Missing value imputation. The missing values were imputed by zero, half of row minimum, row mean, or row median; alternatively, the Python package fancyimpute (version 0.7.0) was used for KNN, IterativeSVD or SoftImpute.

(3) Normalization. No normalization was performed on the protein quantification matrices output by the DIA data analysis software (internal normalization implemented by the label-free quantification algorithms of the software was not taken into consideration in the benchmarking); alternatively, the protein quantification matrices were normalized by sum; the R package limma (version 3.56.2)[52] was used for median normalization; the R package MBQN (version 2.12.0) was used for QN or TRQN. The normalized protein quantification matrices were then $\log_2$ transformed (plus 1 before log transform to avoid log 0).

(4) Batch effect correction. No batch effect correction (NoBC) was performed; alternatively, the R packages limma (removeBatchEffect) or sva (ComBat[53], version 3.48.0) were used; otherwise, the Python package scanorama (version 1.7.4)[54] was used.

(5) Statistical tests. Welch's t-test or Wilcoxon–Mann–Whitney test was performed using the Python package scipy (version 1.5.4); alternatively, the R package limma was used with the trend or voom algorithm; the R package edgeR (version 3.42.4)[55] was used with the QLF or LRT algorithm; the R package DESeq2 (version 1.40.2)[56] was used. The statistical tests were performed mainly based on the $\log_2$ transformed quantities except for limma-voom, DESeq2 and edgeR, which were originally designed for RNA-seq read counts-based data type[36]. For limma-voom, DESeq2, and edgeR, the $\log_2$ quantities after batch effect correction were transformed inversely. Since DESeq2 only accepts integers within the normal read counts range, the quantities were multiplied by 10000 and then rounded up. The raw p-values output by the statistical tests were adjusted by the Benjamini-Hochberg method.

## Metrics for performance evaluation and ranking

For performance evaluation of batch effect correction, the samples were clustered using the Louvain algorithm implemented in the R package Seurat (version 4.3.0.1)[71]. The adjusted Rand index (ARI)[38] metrics were computed by comparing the sample group labels against the clustering results:

$$ARI = \frac{2(ad - bc)}{(a+b)(b+d) + (a+c)(c+d)} \quad (1)$$

where $a$ is the number of pairs with the same true label and in the same cluster, $b$ is the number of pairs with the same true label but in different clusters, $c$ is the number of pairs with the different true labels but in the same cluster, and $d$ is the number of pairs with the different true labels and in different clusters. An ARI of 0 stands for random clustering, and 1 for perfect match.

To evaluate the performance of statistical tests, the adjusted p-value was transformed to $-\log_{10}$ p-value and used as thresholds to classify proteins into positive and negative categories. For human proteins, the ones with $-\log_{10}$ p-value > threshold were false positive (FP) cases and the others were true negative (TN) cases; for yeast proteins, the ones with $-\log_{10}$ p-value > threshold and $\log_2$ fold change (FC) < 0 were true positive (TP) cases; for E. coli proteins, the ones with $-\log_{10}$ p-value > threshold and $\log_2$ FC > 0 were TP cases; the others were false negative (FN) cases. The receiver operator characteristic (ROC) curve was generated using the Python package scikit-learn (version 0.24.2) by plotting true positive rates (TPR) against false

positive rates (FPR) under various thresholds:

$$TPR = \frac{TP}{TP + FN} \quad (2)$$

$$FPR = \frac{FP}{FP + TN} \quad (3)$$

The partial area under the ROC curve (pAUC)[29] was computed within the range of FPR < 10%.

To evaluate the overall performance of differential protein detection, 0.1, 0.05, 0.01, and 0.001 were chosen as p-value thresholds ($p_0$) and the optimized threshold of $|\log_2$ FC| ($t_0$) was searched. For human proteins, the ones with $-\log_{10}$ p-value > $-\log_{10} p_0$ and $|\log_2$ FC | > $t_0$ were FP cases and the others were TN cases; for yeast proteins, the ones with $-\log_{10}$ p-value > $-\log_{10} p_0$ and $\log_2$ FC <$-t_0$ were TP cases; for E. coli proteins, the ones with $-\log_{10}$ p-value > $-\log_{10} p_0$ and $\log_2$ FC > $t_0$ were TP cases; the others were FN cases. The precision–recall (PR) curve was generated using the Python package scikit-learn by plotting precision (also known as positive predictive value, PPV) against recall (i.e., TPR) under various thresholds:

$$Precision = PPV = \frac{TP}{TP + FP} \quad (4)$$

The optimal p-value and $|\log_2$ FC| thresholds were determined to maximize the recall at 95% precision. Other performance metrics, including accuracy values and F1-scores were computed based on the optimal thresholds:

$$Accuracy = \frac{TP + TN}{TP + TN + FP + FN} \quad (5)$$

$$F1 - score = \frac{2 \times Precision \times Recall}{Precision + Recall} \quad (6)$$

The method combinations were ranked by ARI, pAUC, and F1-score separately. A final rank was given by aggregating the ranks of individual metrics:

$$Total\ Rank = Rank\left[Rank(ARI) + Rank(pAUC) + Rank(F1 - score)\right] \quad (7)$$

## Explanation and selection of the high-performing method combinations

To mine the patterns of high-performing method combinations, the benchmarking results were submitted to a learning-to-rank (XGBRanker) model using the Python package xgboost (version 1.5.2)[72]. The method combinations were regarded as categorical features, encoded as an integer array (OrdinalEncoder), and used as input. The ranks were used as output. The objective function was "rank:pairwise", which transformed the ranking task into a pairwise classification problem, learning to predict which item in a pair should be ranked higher. The model assigned an importance score to each feature based on their contribution to the model's predictions. A SHAP explainer (TreeExplainer) was built for the model using the Python package shap (version 0.41.0)[73]. SHAP values (for each method) and SHAP interaction values (for each method pair) were computed for the high-performing (top 25%) method combinations. The mean absolute SHAP values and SHAP interaction values were calculated to measure the global effect of each step and their pairwise influences.

Based on the SHAP explanations, a strategy was established for the recommendations of high-performing method combinations. The steps (i.e., imputation, normalization, batch correction, and statistical test) were ordered by their mean absolute SHAP values. For each step,

candidate method choices were prioritized using their individual contributions. To dynamically refine selections, a beam search algorithm iteratively expanded the method combinations step by step: at each step, it evaluates candidate method choices by combining their standalone contributions with pairwise interaction contributions from previously selected parameters, maintaining only the top-$W$ partial combinations (controlled by beam width $W$; $W = 12$ in this study). In a conservative manner, the 5th percentile ($P_5$) of the SHAP values and SHAP interaction values was used as the standalone contribution of each candidate method choice and pairwise interaction contributions of each method pair (analogous to "value at risk", the maximum loss $-P_5$ expected with a 95% level of confidence). This setting aims to manage the risk of choosing a low-performing method for analyzing a new dataset. This process generated a tree-like structure retaining hierarchical decision rules, where nodes represent method choices and branches indicate conditional priorities. Visualization of the tree was implemented by Graphviz Online (https://dreampuf.github.io/GraphvizOnline/).

### Differential expression analysis and enrichment analysis on real single-cell samples

The selected high-performing method combinations for each DIA analysis software were employed for screening differential proteins between the treatment and control cells (T2Y1E/C2Y1E). For each method combination, differential proteins were determined with adjusted p-value < 0.05 and $|\log_2 \text{FC}| > \log_2 1.5$. GO[61] enrichment was performed using the R package clusterprofiler (version 4.12.6)[74]. Reactome[62] pathway enrichment was performed using the R package ReactomePA (version 1.48.0)[75]. Terms with p-value < 0.05 (Fisher's exact test, adjusted by the Benjamini-Hochberg method) were kept. A hint of overall regulation for each term was given by Z-score calculated based on the numbers of up- and down-regulated proteins[76]:

$$Z = \frac{\text{Count}_{\text{Up}} - \text{Count}_{\text{Down}}}{\sqrt{\text{Count}}} \tag{8}$$

where Z-score > 0 indicates up-regulation and Z-score < 0 for down-regulation. The enrichment result for each method combination was considered as a set of terms with regulation {up/down-regulated term 1, up/down-regulated term 2, …, up/down-regulated term $n$}. Jaccard distance was calculated between each pair of method combinations ($A$, $B$):

$$J(A, B) = \frac{|A \cap B|}{|A \cup B|} = \frac{\left| \left\{ \text{Term } i, |, Z_{Ai} Z_{Bi} \geq 0 \right\} \right|}{|A| + |B| - \left| \left\{ \text{Term } i, |, Z_{Ai} Z_{Bi} \geq 0 \right\} \right|} \tag{9}$$

where $i$ represents any term (independent of regulation) shared by the two method combinations. Specially, Z-score = 0 was considered as either up- or down-regulated. The Jaccard distances were used for hierarchical clustering.

### Reporting summary

Further information on research design is available in the Nature Portfolio Reporting Summary linked to this article.

## Data availability

Raw mass spectrometry data, spectral libraries, and search results have been deposited in the ProteomeXchange Consortium via the iProX[77] partner repository with the dataset identifier PXD056832 or IPX0009767000. Source data are provided with this paper.

## Code availability

The source code of SCPDA is available at Github (https://github.com/WangJianwei1991/SCPDA) and Zenodo (https://zenodo.org/records/17140070)[78].

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

## Acknowledgements

We thank Drs. Mowei Zhou and Mengting Zhang at the Chemistry Instrumentation Center, the Consortium for Advanced Technologies of Mass Spectrometry, Zhejiang University for support and resources for MS data analysis. We thank Shanghai Omicsolution Co., Ltd. for their help in using Spectronaut. We thank Bioinformatics Solutions Inc. for a trial license of PEAKS and guidance in its use. We thank Dr. Dan Zhao at the Department of Chemistry, Fudan University for the kind gift of protein extracts. This work was supported by the National Natural Science Foundation of China (22304153 to Y.Y., 22234007 and 21827806 to Q.F.), and the National Key R&D Program of China (2024YFA1308400 to Y.Y., 2021YFA1301601 to Q.F.).

## Author contributions

J.W. did all the coding work and performed the data analysis. Y.H. performed the majority of the wet-lab experiments. F.L., Z.Y., and Y.J. assisted in the single-cell sample preparation. Q.X. assisted in the MS analysis. S.S. and J.P. provided resources of instrumentation and materials for sample preparation and MS analysis. Y.Y. and Q.F. conceived the project and supervised all aspects of the study. Y.Y. wrote the draft of the manuscript. All the authors revised the paper.

## Competing interests

The authors declare no competing interests.
