## [Transparent Peer Review file · Nature Communications]

Benchmarking informatics workflows for data-independent acquisition single-cell proteomics

Corresponding Author: Dr Yi Yang

Version 0:

Reviewer comments:

Reviewer #1

(Remarks to the Author)

This study by Wang J et al. conducts a comprehensive evaluation of DIA data analysis software tools and workflows for DIA-based single-cell proteomics, which is one of the key issues for current single-cell proteomics studies. They acquired benchmark data from standard hybrid-proteome samples as well as real single-cell samples with spike-in to be processed by different data analysis software and pipelines. Overall, this study provides useful information to guide data mining for single-cell proteomics, yet there are certain limitations in the study design and data presentation/interpretation that need to be addressed.

1. For the hybrid proteome samples, the authors did injection replicates of each sample and these peptides were aliquots taken from digestions of a large amount of protein samples. This design would underestimate the real variations of single-cell proteome samples which are actually prepared from very little amount of proteins subjected to independent digestion. Like most hybrid proteomes used for benchmarking bulk proteomics, the authors could prepare digestion replicates to reflect the inevitable variation introduced in sample prep which is expected to be more significant for single-cell proteomics.

2. When comparing the search results yielded by different search engines or libraries, most comparisons are described without essential details. In line 116-128, for example, it is mentioned 'For DIA-NN (quantification results present in Supplementary Data 1), the public spectral library-based strategy quantified more proteins and peptides, while it resulted in lower quantitative accuracy than the other data analysis strategies (Supplementary Note 2, as well as Supplementary Figs. 1–6). The library-free workflow yielded high data completeness and good protein quantitative accuracy.' But what is the exact difference? Is this difference significant enough? In Supplementary Figs. 1 and 2, different libraries used by DIA-NN seem to generate very similar results in terms of quantification accuracy and reproducibility. It's the same for line 151-165 and other sections which lack critical data and only describe a rough trend in comparison.

3. In this paragraph (line 116-128) and throughout the manuscript, the authors support specific statements with multiple supplementary figures. A statement is better to be supported by citing one or two figure panels illustrating the central point. The current manuscript is somehow weakened by a large number of supplementary materials and notes (some notes are identical with the text) which can be cut-down and re-organized to explicitly and concisely deliver the messages.

4. When analyzing results from comparing various data processing steps for differential expression analysis, the data presentation could be more informative and concise. For example, Figure 2f is sort of difficult to read with small fonts and many combinations. Can the authors just list a few key metrics (e.g. ARI, total DEP number, false positive rate with the general criteria $FC > 1.5$ and $p < 0.05$) for top3 best combination and last3 worst combination? In this way, it may be easier to know which steps would affect the DEP analysis result to what extent. Are Figs 2e and 3d based on DEP analysis with the regular criteria of $FC > 1.5$ and $p < 0.05$? This should be specified in the text and figure legend. As also noted in this study that 'differential proteins are typically detected at a given fold change (FC) and p-value threshold rather than an FPR range', results shown in Figs 2e and 3d are more informative if the combination of different tools and steps are specified.

5. The same problem exists when evaluating DEP analysis by Spectronaut or PEAKs (line 276-279). A panel of supplementary figures and notes are provided and they could be simplified.

6. When evaluating informatics workflows for analyzing real single-cell proteomes with spike-in digests, several optimal steps (such as the normalization method) differ from those identified from the standard hybrid proteome benchmark. Does fig. 3d indicate that sparsity reduction is no longer useful for false-positive control in real single-cell proteome analysis? This key point is not mentioned or discussed in the manuscript, and seems contradictory to the author recommendation in Discussion that states 'sparsity reduction is a primacy in the whole workflow, where 75% data completeness is a good trade-off...'. Also figure 3e is better to be re-formatted to focus on major evaluation metrics.

Minor points:

Line 160-165, no figures are cited to support the statements.

Line 191, 'Under constraints of 75% data completeness, the false positive hits were <5% at the protein level and ~1% at the peptide level by DIA-NN and Spectronaut.' Here should mention, by applying the additional filtering step, what % of the initial protein identification number (at 50% data completeness) is lost for the two software tools.

Line 313, 'limma-voom and limma-trend were robust with the three mainstream DIA data analysis software for single cell proteomics (Fig. 3f, as well as Supplementary Figs. 41, 45, and 49).' Here Fig. 3f seems missing.

(Remarks on code availability)

Reviewer #2

(Remarks to the Author)

I believe this is a very comprehensive work of sufficient interest for Nature Communications. The benchmarks shown on Figure 3 are of particular interest and are highly useful in understanding the factors influencing quantification in single cell proteomics. I can therefore recommend publication provided the below comments are addressed.

- It needs to be clear how the output of each software tool was processed and what filtering was applied in each benchmark, currently this is not specified.
- Charge 1 precursors must not be considered when analyzing dia-PASEF data.
- Human peptide and protein identifications should not be counted as false due to the ubiquitous presence of common contaminants. Respectively, calculations on lines 192-193 are not relevant.
- 194-195 – not correct, the authors count only known false positives and do not estimate the total false positive rate. This is fine, but the conclusion on the lines 194-195 cannot be made.
- DIA-NN and, if possible, Spectronaut and PEAKS should be updated to the recent versions to increase the relevance of this work.
- Given that statistical tests were performed on log-transformed quantities, how was it possible to impute missing values with 0? Whenever allowed by the statistical test used, the authors need to compare to no imputation. If not compatible with the test, the authors need to replace the test with a different one that accepts missing values.
- Only row-based imputation methods can be considered reliable for DE analysis, i.e. if the other methods are also tested for DE, their potential to produce false positives needs to be discussed.
- What was the difference in LC-MS state/settings between the batches? Are any proteins differentially expressed between the batches or the variation between batches is negligible and does not model batch effects in real experiments?
- Please use PCA instead or in addition to UMAP for visualizing batches, UMAP interpretation is highly problematic.
- Plots like on Figure 2f are difficult to interpret, I recommend authors either remove them or visualize differently.
- The authors should establish the robustness of the ARI and purity metrics at least by (i) comparing the metric values between the cases when different individual runs are excluded from consideration and, preferably, also by (ii) splitting the set of runs in two groups, randomly but in a balanced fashion, and evaluating the degree of similarity of the metrics calculated separately for each group.
- The authors should discuss whether clustering-based metrics such as ARI and purity are useful in practice. Missing values and imputation are the major factors affecting clustering, however they do not have an effect on the detection of proteins with complete profiles as DE. I believe the number of true/false DE proteins is a significantly more reliable metric and should be used as a basis for imputation, normalisation and batch correction methods comparison instead or in addition to ARI/purity.

(Remarks on code availability)

Reviewer #3

(Remarks to the Author)

In their manuscript "Benchmarking informatics workflows for data-independent acquisition single-cell proteomics" Wang et al. present a comprehensive benchmark of data analysis workflows for single-cell proteomics. Using a simulated sample consisting of mixed proteomes and real single-cell samples with a spike-in scheme, they benchmark different steps during data analysis including DIA data analysis software, sparsity reduction, missing value imputation, normalization, batch effect correction, and differential expression analysis. The manuscript is well structured and presented. I agree with the use of

mixed species samples, however I have some concerns regarding the data analysis. The benchmarks using the real single-cell samples are a good idea to include chemical noise to the analysis, however I have concerns about the way the data is analyzed.

Major points:

1.
Line 49: Different DIA data analysis solutions have indeed been compared previously in <https://doi.org/10.1038/s41467-024-52605-x> and <https://doi.org/10.1016/j.mcpro.2024.100839>
The authors should thus tone down their statements accordingly and cite these papers.
2.
To compare quantitative accuracy of their workflows, the authors use violin plots that describe the distribution of measured fold-changes. Then they compare the deviation of the median of these distributions to the true fold-change to highlight the best method. These comparisons are very close throughout the manuscript and thus, mostly not very conclusive. Unfortunately, the median does not measure the spread of the fold-changes, which is quite important. Furthermore, many of the comparisons between search-engines could mainly be driven by the different number of precursors and the non-overlapping precursors. For example, if more low-abundant precursors are identified, these could decrease the overall accuracy although the overlapping precursors are equally well quantified.
The authors should compare quantitative accuracy using a measure that includes the spread of the fold-changes. E.g. the distributions of the absolute fold-change errors. Furthermore, the authors should make sure that the differences between search engines are not produced by differences in the number of precursors or the non-overlapping precursors by repeating their comparison with only overlapping precursors.
3.
In Line 182, the authors show very high fractions of false transfers to E. coli data from human and yeast. Can the authors rule out the possibility that these peptides are indeed present in the sample due to carry-over on the liquid chromatography column from previous samples containing human and yeast?
4.
For the batch correction, did the authors provide biological covariates to the methods? For Combat that would be the model matrix where one can specify the outcome of interest. As far as I know, Scanorama does not use biological covariates, which would explain its lower performance. If the authors provided biological covariates, it would be important to point out this difference and show how Combat and limma perform if no biological covariates are provided to the model. These covariates are not always available in single-cell data.
5.
Regarding the differential expression benchmarks. A common strategy in proteomics is to ignore missing values for DE testing. Did the authors consider including that into the comparisons?
6.
Lines 597 & 639: Did the authors use the same number of clusters across comparisons to calculate ARI? How stable are these results across different numbers of clusters?
7.
For the real single-cell dataset, the most relevant normalization method should be no-normalization, because the ratio spike-in is constant and only the single-cell background is variable. All the applied normalization methods are not compatible with these samples, because they try to align distributions that are different by experimental design. The ratios will be distorted by the size of the single cell; however, the size of the single-cell is not relevant for this experiment. Maybe the batch-correction step corrects for many of these un-suitable normalizations that were applied.
Apart from no-normalization, one could normalize only based on the spike-in peptides, which are assumed to have the same total amounts across samples. Thus, sum-normalization should perform well here. Going even further, all processing steps should only be applied on a matrix containing only the spike-in peptides, because also the batch-correction will work with the assumption that the real distributions of the samples are comparable, which they are not due to the variable single-cell sizes. The single-cells should only provide the chemical background that challenges the overall analysis.

Minor points:

1.
Lines 23, 82 & 414: this should say "simulated" instead of "stimulated".
2.
Fig. 1b: From the figure or its caption, it's not clear that this is from the library-free data.
3.
Lines 131-133 – the authors claim that PEAKS quantified more analytes than Spectronaut, but then state that PEAKS quantified 11,165 peptides vs 12,537 peptides in Spectronaut. With peptides being the analyte measured in LC-MS based proteomics, I would argue it is the opposite?
4.
Line 202: How did the authors create these different batches?
5.
Line 433: Working with 75% data completeness is possible for the authors only because they have very similar samples at hand (ie. homogeneous cell lines). I think it would be important to note that this will be significantly more complicated when analyzing many different cell types, where missing values will be much higher due to cell-type specific effects.
6.
Line 573: Did the authors deactivate the internal normalization of the search engines or did they use the standard settings? This is important for RT-dependent normalizations, as in DIA-NN this cannot be reversed.

(Remarks on code availability)

Version 1:

Reviewer comments:

Reviewer #1

(Remarks to the Author)

In the revision, the authors have made great efforts to improve the manuscript. Only a few minor points remain to be addressed and the reviewer would recommend publication of this pretty comprehensive work after the very minor revision.

Line 147: Spectronaut should be DIA-NN.

In Figure 4C annotation, does the blue bar represent TP and the green is for TN?

Line 454: For this key statement "When a project specific spectral library is not available (e.g., the samples are rare or resources for additional injections are not sufficient), the built-in prediction by DIA-NN is preferred for library-free analysis." it seems contradictory to what are shown in Results. Spectronaut in directDIA yields more protein IDs, higher data completeness and comparable FPRs in differential protein analysis (Figs 1B, 1C, and 4C). So would this point be mentioned and recommended in Discussion?

(Remarks on code availability)

Reviewer #2

(Remarks to the Author)

The authors have revised and significantly improved the manuscript, addressing the most important comments. As noted before, this is an excellent study that should be published, provided several easy to deal with issues are addressed.

The authors now clarify the software settings that they have applied in their analysis. However, there are two apparent issues. While the authors state protein FDR was controlled at 1%, the pg matrix output of DIA-NN that does not provide protein FDR control was used. How did authors achieve protein FDR control? For peptide-level quantification, the authors averaged precursor quantities. This is unlikely to produce quality results, especially in the context of high missing value rates inherent to SCP. Can the authors use MaxLFQ or Top N method instead?

I noted that human identifications should not be counted as false in FDR tests. The authors argue that removing peptides found in blanks from consideration makes this acceptable. This is incorrect for a number of reasons, one being that the blanks cannot be reliably assumed to represent the degree of contaminant presence in SCP samples. Given that this analysis is not just uninterpretable for the aforementioned reason but also completely unnecessary here, as using non-human entrapment identifications is sufficient, it should be removed.

(Remarks on code availability)

Reviewer #3

(Remarks to the Author)

I have received this manuscript for review only after the first round of revisions, and had a thorough look at the rebuttal and updated manuscript. I can confirm that my initial concerns were shared by the other reviewers, and have now been appropriately addressed by the authors. I think the manuscript serves as a useful resource for the field, and especially appreciate the use of real single-cell samples as this is critical for establishing 'best practises' in the field.

Therefore, I deem the manuscript suitable for publication in its current state.

(Remarks on code availability)

Version 2:

Reviewer comments:

Reviewer #2

(Remarks to the Author)

First, I apologise for a huge delay with the review, this happened due to technical issues with my email.

- The new software settings description is self-contradictory. "The Q-value cut-off at both precursor and protein level was 1%" contradicts for DIA-NN "PG.Q.Value at 0.05" and for Spectronaut "Protein Qvalue Cutoff (Run): 0.05" in Supp Table 6. The settings used are sensible considering the focus of the study on single cells, however the authors need to discuss the choice of 5% Q-value cut-off, to highlight this for the reader. Further, the FDR benchmarks, in particular on SF8, need to be considered in the context of this cut-off, which needs to be made clear so that the reader does not jump to wrong conclusions about the overall FDR of the softwares.

- The authors should not blindly agree with reviewer 1, if such an agreement contradicts the actual data when it comes to software comparisons. In the benchmarking by the authors, Spectronaut seems to produce higher protein numbers and completeness but at the cost of higher FDR (SF8), this makes the comparison of numbers and completeness between the softwares inconclusive. In turn, the TPR, FPR, TN and FN numbers vary across comparisons and likewise make definitive conclusions impossible. The authors therefore should not summarise their findings in the form "software A is better than software B in scenario X". This in fact masks the true impact of the present work, which in contrast shows a complex picture with different workflows having distinct advantages and drawbacks. For example, DIA-NN demonstrates by far superior TPR and TN on figure 4c, robustly across methods, while Spectronaut's performance in the same benchmark curiously is sensitive to the method choice. And on SF14d DIA-NN has higher NoSR FPR for S4/S2 than Spectronaut but lower for S5/S1. Such details are valuable to the reader and should not be summarised as 'recommendations'. These therefore need to be removed. However I can support the recommendation to analyze with specific libraries and keep track of the data completeness, based on the data shown in the manuscript. The analysis of statistical workflows is also excellent and is the true strength of this work.

- The risk of co-searching is exaggerated. The reason the authors observe higher FPR for co-search (SN8) is partially because q-value filter was set to 5% on run level and partially because the relationship between FPR and FDR, that is controlled by the softwares, is different depending on whether samples are searched together or separately.

(Remarks on code availability)

Responses to the Reviewer's Comments

Benchmarking informatics workflows for data-independent acquisition
single-cell proteomics

Jianwei Wang et al.

Revision Summary

During the revision, we have mainly:

- (1) Performed additional experiment to mimic the real variations of single-cell proteome samples which are actually prepared from very little amount of proteins subjected to independent digestion.
- (2) Reanalyzed the data using the latest versions of software that we had access when we started the revision. Most of the results were updated.
- (3) Reorganized the presentation and visualization. The supplementary materials were cut down, where only the Supplementary Figs. illustrating the central point were kept and the others were downgraded into Supplementary Data.
- (4) Further mined the benchmarking results. A learning-to-rank model and SHAP explanations to discover the patterns in the high-performing method combinations. Based on the patterns, we established a strategy for the recommendation of method combinations and evaluated its generalizability in characterizing real single cell proteome samples. We believe this can facilitate the utility of our strategy in the field.

All the corresponding changes have been highlighted in the marked revised manuscript. Our point-by-point responses to each reviewer's comments are present below.

Reviewer #1:

This study by Wang J et al. conducts a comprehensive evaluation of DIA data analysis software tools and workflows for DIA-based single-cell proteomics, which is one of the key issues for current single-cell proteomics studies. They acquired benchmark data from standard hybrid-proteome samples as well as real single-cell samples with spike-in to be processed by different data analysis software and pipelines. Overall, this study provides useful information to guide data mining for single-cell proteomics, yet there are certain limitations in the study design and data presentation/interpretation that need to be addressed.

Response: Thank you very much for taking the time and effort to review our manuscript. We really appreciate the valuable comments and suggestions, and have modified our manuscript accordingly.

1. For the hybrid proteome samples, the authors did injection replicates of each sample and these peptides were aliquots taken from digestions of a large amount of protein samples. This design would underestimate the real variations of single-cell proteome samples which are actually prepared from very little amount of proteins subjected to independent digestion. Like most hybrid proteomes used for benchmarking bulk proteomics, the authors could prepare digestion replicates to reflect the inevitable variation introduced in sample prep which is expected to be more significant for single-cell proteomics.

Response: Thank you for the suggestions. We totally agree the significance to reflect the variation introduced in sample preparation. In the revised version, we performed another experiment to mimic the real variations of single-cell proteome samples which are actually prepared from very little amount of proteins subjected to independent digestion (250 pg proteins in total as starting materials). The main thread of the revised manuscript is: in the first part, the design of benchmarking on data of injection replicates aims at focusing on the technical performance of each software evaluated on ground-truth samples. With the benchmarking framework established, the dataset of proteome samples prepared with independent digestion was used to evaluate if the informatic workflows cope with the real variations of single-cell proteome samples.

Corresponding revision was made on pages 8 and 13 of the main text. Performance of different software and searching strategies on the independent-digestion dataset is summarized in **Supplementary Data 2**. Benchmarking results of informatic workflows are present in **Fig. 3, Supplementary Figs. 18 and 19**, as well as **Supplementary Data 4**.

2. When comparing the search results yielded by different search engines or libraries, most comparisons are described without essential details. In line 116-128, for example, it is mentioned ‘For DIA-NN (quantification results present in Supplementary Data 1), the public spectral library-based strategy quantified more proteins and peptides, while it resulted in lower quantitative accuracy than the other data analysis strategies (Supplementary Note 2, as well as Supplementary Figs. 1–6). The library-free workflow yielded high data completeness and good protein quantitative accuracy.’ But what is the exact difference? Is this difference significant enough? In Supplementary Figs. 1 and 2, different libraries used by DIA-NN seem to generate very similar results in terms of quantification accuracy and reproducibility. It’s the same for line 151-165 and other sections which lack critical data and only describe a rough trend in comparison.

Response: Thank you for the question. We are sorry for the confusing expression in the previous manuscript. In the revised version, we clarified the thread: First, we compared the performance of these searching strategies within each software. Due to the length limitation, we stated only the conclusions in the main text and placed the details in **Supplementary Note 2**. Next, considering the potential limitation of spectral library availability in practical applications, we focused on the inter-software performance comparison without the need of external spectral libraries. Detailed comparison results are present in the main text. Then, we stated the conclusions in brief of inter-software performance comparison with other searching strategies.

For the convenience of a reader, critical data of performance comparison is summarized in **Supplementary Data 1**. Corresponding revision was made on pages 6 and 7 of the main text.

In addition, significance of the differences was evaluated by statistical tests (t-test p-value < 0.05 for the numbers of quantified proteins/peptides; t-test p-value < 0.05 and Cohen’s $|d| > 0.2$ for the quantification accuracy). **Figure 1** and the related **Supplementary Figs.** were updated to include the p-values.

3. In this paragraph (line 116-128) and throughout the manuscript, the authors support specific statements with multiple supple figures. A statement is better to be supported by citing one or two figure panels illustrating the central point. The current manuscript is somehow weakened by a large number of supple materials and notes (some notes are identical with the text) which can be cut-down and re-organized to explicitly and concisely deliver the messages.

Response: Thank you for the suggestions. As explained in the response to the previous comment, we have to place the details in **Supplementary Note 2** due to the length limitation. However, in the revised version, we re-organized the supplementary materials. Supplementary Notes with similar text were cut down and combined (into **Supplementary Note 2**), which cites the related figure panels. Only the Supplementary Figs. illustrating the central point were kept and the others were downgraded into Supplementary Data.

Supplementary Figure 1. Performance comparison of different searching strategies using DIA-NN at the protein level.

a Numbers of quantified proteins per run. The bars indicate the mean values and the error bars indicate the standard deviations. Significant differences (t-test p-value < 0.05) are indicated. **b** Numbers of proteins quantified in at least specified percentages (data completeness) of runs. **c** Overlap of the proteins quantified in at least 50% runs. **d** Distribution of the coefficient of variation (CV). CV values were calculated only for proteins quantified in at least 3 runs per sample. The median values are indicated. **e** Measured fold change (FC) values of protein quantities using sample S3 as reference. FC values were calculated only for proteins quantified in at least 3 runs for each sample of the comparison. Numbers (n) of proteins are indicated for each species. The boxes mark the first and third quartile and the lines inside the boxes mark the median; the whiskers extend from the box to the farthest point lying within 1.5 times the inter-quartile range; outliers are not shown. The theoretical ratios are highlighted as dashed lines. Differences between the measured median FC values and theoretical values are indicated, among which the smallest ones are darkened. Significant differences (t-test p-value < 0.05 and Cohen's $|d| > 0.2$) are indicated. Source data are provided as a Source Data file.

4. When analyzing results from comparing various data processing steps for differential expression analysis, the data presentation could be more informative and concise. For example, Figure 2f is sort of difficult to read with small fonts and many combinations. Can the authors just list a few key metrics (e.g. ARI, total DEP number, false positive rate with the general criteria $FC > 1.5$ and $p < 0.05$) for top3 best combination and last3 worst combination? In this way, it may be easier to know which steps would affect the DEP analysis result to what extent.

Are Figs 2e and 3d based on DEP analysis with the regular criteria of $FC > 1.5$ and $p < 0.05$? This should be specified in the text and figure legend. As also noted in this study that ‘differential proteins are typically detected at a given fold change (FC) and p-value threshold rather than an FPR range’, results shown in Figs 2e and 3d are more informative if the combination of different tools and steps are specified.

Response: Thank you for the suggestions. We are sorry for the confusing presentation in the previous manuscript. In the revised version, the figures were re-organized. In **Fig. 2b**, the parallel coordinate representation shows metrics using different method combinations, where the number of highlighted method combinations are reduced to top 5% and the top 3 are overstriking. In **Figs. 2c** and **2d**, the metrics are visualized in a hyperbox, the method choice in each step of the best method combination is marked with dashed lines.

Each step of the best method combination and the criteria for differential analysis are labeled in the figure. Since the actual fold change (FC) between the samples S4 and S2 is 1.5, the regular criteria of $|\log_2 FC| > \log_2 1.5$ may be too strict. We used $|\log_2 FC| > \log_2 1.2$ for S4/S2 and $|\log_2 FC| > \log_2 1.4$ for S5/S1, which was determined by a preliminary survey (**Supplementary Note 6**).

In order to show which steps would affect the result to what extent, we further fitted the results using a learning-to-rank model and SHAP explanations (**Supplementary Fig. 17**). The resulting SHAP values quantify the contribution of each feature (method) to a prediction, and the SHAP interaction values extend this by measuring how pairs of features (methods of two steps) jointly influence predictions, capturing their combined effects beyond individual contributions. Corresponding revision was made on page 12 of the main text.

We expect that with these modifications, the data presentation has been more informative and concise.

S4/S2 SR75 p-value < 0.05 $|\log_2 FC| > \log_2 1.2$

Supplementary Figure 17. Explanations of the patterns of high-performing method combinations (S4/S2 SR75) **a** Feature importance of the model. **b** Mean absolute SHAP values for each step. **c** SHAP values for the method choices in each step. **d** Mean absolute SHAP interaction values for each two steps. **e** SHAP interaction values for pairwise method choices in each two steps. In **c** and **e**, the boxes mark the first and third quartile and the lines inside the boxes mark the median; the whiskers extend from the box to the farthest point lying within 1.5 times the inter-quartile range. Individual data points are overlaid as dots. The median values and frequency (n) are indicated for each method choice. The data are processed starting with SR75. Differential analysis was performed between the S4 and S2 sample groups. Differential proteins are determined with p-value < 0.05 and $|\log_2 FC| > \log_2 1.2$. Only the top 25% method combinations are subjected to SHAP explanation and the method choices with $n < 3$ are not shown. Source data are provided as a Source Data file.

5. The same problem exists when evaluating DEP analysis by Spectronaut or PEAKs (line 276-279). A panel of supple figures and notes are provided and they could be simplified.

Response: Thank you for the suggestions. As explained in the response to the previous comment, we re-organized the figures. The related figure panels were cited. Corresponding revision was made on page 12 of the main text.

6. When evaluating informatics workflows for analyzing real single-cell proteomes with spike-in digests, several optimal steps (such as the normalization method) differ from those identified from the standard hybrid proteome benchmark. Does fig. 3d indicate that sparsity reduction is no longer useful for false-positive control in real single-cell proteome analysis? This key point is not mentioned or discussed in the manuscript, and seems contradictory to the author recommendation in Discussion that states ‘sparsity reduction is a primacy in the whole workflow, where 75% data completeness is a good trade-off...’.

Also figure 3e is better to be re-formatted to focus on major evaluation metrics.

Response: Thank you for the suggestions. In the previous version of manuscript, the abundance of human proteins in the real single-cell samples may not be constant due to the cell heterogeneity. Thus, the count of false positives may be inaccurate.

In the revised version, we analyzed the mixed-organism samples mimicking the real variations of single-cell proteome samples with independent digestion, where the human proteins should be considered negative. The result demonstrated that 75% sparsity reduction is still useful for false positive control (Figure SD4-26 in **Supplementary Data 4**).

In addition, to avoid the distortion of ratios by the size of the single cells when analyzing the real single-cell samples with spike-in digests, we selected a subset of human proteins, including histones with cell size-independent copy numbers (*Molecular & Cellular Proteomics*, 2014, 13: 3497–506, DOI:10.1074/mcp.M113.037309; *Journal Proteome Research*, 2023, 22: 3773–3779, DOI:10.1021/acs.jproteome.3c00441), as well as housekeeping proteins with probably stable expression (*Trends in Genetics*, 2013, 29: 569–574, DOI:10.1016/j.tig.2013.05.010; *GigaScience*, 2019, 8: giz106, DOI:10.1093/gigascience/giz106) for the control groups. All the processing steps were applied on a matrix containing only the spike-ins and the subset of human proteins. In this setting, other proteins from the single cells should only provide the chemical background that challenges the overall analysis. False positives were at a low level for different sparsity reduction criteria (**Supplementary Figs. 20**).

Also, the figures have been re-formatted to focus on major evaluation metrics.

Figure SD4-26. Comparison of results with different software and sparsity reduction conditions

a Correlations of the ranks and metrics of the method combinations between results obtained under varying sparsity reduction conditions (0%, 66%, 75%, 90%), holding constant both the comparison group and the p-value threshold. **b** Correlations between the comparison groups (S4/S2 and S5/S1), under identical sparsity reduction conditions and p-value thresholds. **c** Correlations between results by different software. In **a–c**, the boxes mark the first and third quartile and the lines inside the boxes mark the median; the whiskers extend from the box to the farthest point lying within 1.5 times the inter-quartile range. Individual data points are overlaid as dots. The median values are indicated. **d** Numbers of detected proteins using different sparsity reduction conditions. The top ranked method combination starting with each sparsity reduction condition is shown, whose serial number is in brackets. Mappings of the serial numbers to detailed method combinations are present in Fig. 2a. The blue bars indicate the true negative (TN) proteins, the green bars indicate the true positive (TP) cases, the red bars indicate the false positive (FP) cases, and the purple bars indicate the false negative (FN) cases. In **c** and **d**, differential proteins are determined with p-value < 0.05.

Supplementary Figure 20. Comparison of results with different software and sparsity reduction conditions on the spike-in single-cell samples

Performance of the selected high-performing method combinations on the spike-in single-cell samples. Metrics include: the ARI and pAUC values (indicated by dot sizes and colors), numbers of detected TN (blue bars), TP (green bars), FP (red bars), and FN (purple bars) proteins, as well as TPR (green lines) and FPR (red lines) values. The top ranked method combination starting with each sparsity reduction condition is shown, whose serial number is in brackets. Mappings of the serial numbers to detailed methods for each step are present in Fig 2a. The data were processed starting from SR75. Differential analysis was performed between the C2Y1E and C1Y2E sample groups. Differential proteins are determined with $p\text{-value} < 0.05$ and $|\log_2 FC| > \log_2 1.5$.

Minor points:

Line 160-165, no figures are cited to support the statements.

Response: Thank you for the comment. In the revised version, we re-organized the supplementary materials and critical data to support the statements is summarized in **Supplementary Data 1**.

Line 191, 'Under constraints of 75% data completeness, the false positive hits were <5% at the protein level and ~1% at the peptide level by DIA-NN and Spectronaut.' Here should mention,

by applying the additional filtering step, what % of the initial protein identification number (at 50% data completeness) is lost for the two software tools.

Response: Thank you for the comment. In the revised version, we changed the method of error rate estimation and the data were updated. We mentioned both the percentage of removed entrapment hits and the loss of the sample-specific proteins by applying the filtering step. Corresponding revision was made on page 9 of the main text.

Line 313, 'limma-voom and limma-trend were robust with the three mainstream DIA data analysis software for single cell proteomics (Fig. 3f, as well as Supplementary Figs. 41, 45, and 49).' Here Fig. 3f seems missing.

Response: Thank you for the comment. We are sorry for the mistake in the previous manuscript. In the revised version, the figures were re-organized and this statement was removed.

Reviewer #2:

I believe this is a very comprehensive work of sufficient interest for Nature Communications. The benchmarks shown on Figure 3 are of particular interest and are highly useful in understanding the factors influencing quantification in single cell proteomics. I can therefore recommend publication provided the below comments are addressed.

Response: Thank you very much for taking the time and effort to review our manuscript. We really appreciate the valuable comments and suggestions, and have modified our manuscript accordingly.

- It needs to be clear how the output of each software tool was processed and what filtering was applied in each benchmark, currently this is not specified.

Response: Thank you for the comment. For DIA-NN, the output quantification matrix files (pg_matrix.tsv and pr_matrix.tsv) were used for benchmarking. For the peptide-level quantification, the precursor matrix was aggregated by “Stripped.Sequence” and the quantity of each peptide was the sum of those of the corresponding precursors.

For Spectronaut, the exported report files were used for benchmarking. For the protein-level quantification, the columns “PG.ProteinAccessions” and “PG.Quantity” were used. For the peptide-level quantification, the columns “PEP.StrippedSequence” and “PEP.Quantity” were used.

For PEAKS, the exported quantification report files (lfq.dia.proteins.csv and lfq.dia.peptides.csv) were used for benchmarking. For the protein-level quantification, the columns “Accession” and “Area [SampleName]” were used, where only the top protein in each protein group (indicated by the columns “Top” and “Protein Group”) were kept. For the peptide-level quantification, the columns “Peptide” and “Area [SampleName]” were used. For the error rate estimation with single-organism samples, the exported identification report files (dia_db.proteins.csv and dia_db.peptides.csv for library-free analyses, or sl.proteins.csv and sl.peptides.csv for spectral library searches) were used.

Corresponding revision was made in the Methods section on page 24 and 25 of the main text.

- Charge 1 precursors must not be considered when analyzing dia-PASEF data.

Response: Thank you for the comment. We agree that charge 1+ precursors should be excluded in analyzing diaPASEF results. During the analyses, all of the software workflows were run using the default settings recommended for diaPASEF data with minimized modifications to make their results comparable. We checked the results and there were few charge 1+ precursors (<0.1%). They were negligible in the quantification matrices and thus would not have impact on the benchmarking results.

- Human peptide and protein identifications should not be counted as false due to the ubiquitous presence of common contaminants. Respectively, calculations on lines 192-193 are not relevant.

Response: Thank you for the comment. Considering the identified entrapment analytes that were originated from potential common contaminants, blank samples were inserted among the single-organism samples. For the peptide standard samples, 3 blank LC-MS injections were performed; for the independent-digestion samples, 3 blank samples were subjected to digestion before LC-MS analysis and another 3 blank LC-MS injections were performed. The entrapment analytes identified from the blank samples were fewer than those from the single-organism samples, indicating the latter (at least a large proportion of them) were not among the contaminants and not present in the sample (**Supplementary Figs. 8a and 9a**).

To calculate the error rates, the numbers of the sample-specific and entrapment analytes were adjusted by considering the potential contaminants identified in the blank samples. Proteins detected in the blank samples using any software and searching strategies were combined to a contaminant set. For each run, the sample-specific analytes were the organism-specific analytes added with the wrong-organism analytes present in the contaminant set. The entrapment analytes were the wrong-organism analytes not present in the contaminant set.

Corresponding revision was made in **Supplementary Note 3**.

Supplementary Figure 8. Comparison of false positive detection by different software using the library free searching strategy at the protein level.

a Numbers of quantified proteins per run. For each sample, correctly detected proteins should be from the organism specific to the sample (in green), while those from other organisms (in red) are potential false positives. Results of blank injections are shown to assess potential contaminants. **b** Numbers of organism-matched (in green) and potential false positive (in red) proteins quantified in at least specified percentages (data completeness) of runs.

- 194-195 – not correct, the authors count only known false positives and do not estimate the total false positive rate. This is fine, but the conclusion on the lines 194-195 cannot be made.

Response: Thank you for the comment. We agree that the percentage of entrapment hits reported in the previous version of manuscript was not exactly the true error rate. In the revised version, estimation of error rates was conducted referring to the paper of DIA-NN (*Nature Communications*, 2022, 13: 3944, DOI:10.1038/s41467-022-31492-0). The π_0 correction factor (the percentage of incorrect targets) will allow the estimated error rate including not only the known false positives (entrapment) but also those in the targets.

The error rate was estimated for each sample group as

$$\text{Error rate} = \frac{E_{id}}{E_{id} + S_{id}} \cdot \frac{E_{lib} + S_{lib}}{E_{lib}} \cdot \pi_0 \quad (\text{S3-1})$$

where S_{id} is the number of identified organism-specific proteins or peptides for each sample, and E_{id} is the number of identified entrapment proteins or peptides that are not expected to be present in the sample. S_{lib} and E_{lib} are the respective numbers of organism-specific and entrapment proteins peptides in the spectral library. The π_0 (prior probability of incorrect identification) correction factor was calculated as

$$\pi_0 = \frac{E_{lib} + S_{lib} - 0.99 S_{id}}{E_{lib} + S_{lib}} \quad (\text{S3-2})$$

where the factor is set as 0.99 since S_{id} is the number of identified organism-specific proteins or peptides at 1% q-value.

In the revised version, we focus on the comparison of co-searches and separate searches relatively, rather than the exact absolute error rates of different software. The original conclusion was removed.

The calculation process and results of the error rates are present in **Supplementary Data 1** and **2**. Corresponding revision was made on page 8 of the main text and **Supplementary Note 3**.

It should be noted that the term “false positive rate” (FPR) is false positive / (false positive + true negative) in a classification task. This is a different concept of error rate, false positive / (false positive + true positive). We report error rates in the section of performance comparison of DIA data analysis, and FPR in the section of benchmarking the workflows for differential analyses. We have differentiated them as far as possible to avoid confusion.

- DIA-NN and, if possible, Spectronaut and PEAKS should be updated to the recent versions to increase the relevance of this work.

Response: Thank you for the comment. In the revised version, we reanalyzed the data by DIA-NN (version 1.9.2) and Spectronaut (version 19.5), which were the latest versions that we had access when we started the revision.

As the software is rapidly evolving, benchmarking of their performance need active update. This calls for sustainable efforts beyond this study by the community. We released the framework for benchmarking and analysis as an open access tool. Corresponding discussion was made on page 19 of the main text.

- Given that statistical tests were performed on log-transformed quantities, how was it possible to impute missing values with 0? Whenever allowed by the statistical test used, the authors need to compare to no imputation. If not compatible with the test, the authors need to replace the test with a different one that accepts missing values.

Response: Thank you for the comments. The transform performed on the quantities was plus 1 before log transform to avoid log 0, which has been clarified in the Methods section (on page 25).

In the revised version, we tested the performance with no imputation whenever allowed by the batch correction and statistical test methods used. In these benchmarking workflows, missing values were kept as they were (KeepNA). Batch correction methods included NoBC and limma, and statistical test methods included t-test, Wilcox, and limma-trend. Other methods were excluded since they do not support missing values.

To compare the performance with and without imputation, benchmarking results with KeepNA were ranked combined with the original ones. We focused on the results of S4/S2 with SR75. The top 1% method combinations with and without imputation are visualized in **Supplementary Data 3** (Figures SD3-46 for DIA-NN, SD3-47 for Spectronaut, SD3-48 for PEAKS), showing the gap between them. The low performance of the method combinations without imputation may be due to the limited choices of batch correction and statistical test methods. Other the other hand, with the method choices in other steps fixed, the benchmarking metrics without imputation were still lower than those with imputation.

Corresponding revision was made on page 12 of the main text and **Supplementary Note 10**.

(Legend on next page)

Figure SD3-46. Performance comparison of method combinations with and without imputation (DIA-NN S4/S2 SR75)

a The evaluated method combinations. Method choices in blue can process data without imputation (containing missing values) whereas those in gray cannot. **b** Parallel coordinate representation showing metrics using different method combinations with or without imputation, ranked together. **c** Compositions of the top 1% and 5% method combinations in **b**. Mappings of the serial numbers to detailed methods for each step are present in **a**. **d** Adjusted Rand index (ARI) metrics. **e** Partial area under receiver operator characteristic curve (pAUC) metrics. In **d** and **e**, the metrics are visualized in a hyperbox, where each face displays the metrics with two steps variable and the other steps fixed to those of the best method combination without imputation. For the best method combination, the method choice in each step is marked with dashed lines. Dot sizes and colors indicate the metric values. **f** Accuracy (dots), recall (triangles), precision (squares), and F1-score (bars) metrics. Rows represent batch effect correction methods and columns represent normalization methods. The other steps are those of the best method combination. **g** Clustering result of the 5 groups of samples visualized using principal component analysis for dimension reduction. The fill colors indicate the sample groups and the shape indicate the batches. The border colors indicate the clusters. **h** Receiver operator characteristic (ROC) curves using $-\log_{10}$ p-value as scores. The optimal cut-offs with false positive rate (FPR) ≤ 0.1 are marked using black dots with score threshold (Thr), FPR, and true positive rate (TPR) values indicated. **i** Volcano plots. Blue dots represent TP *E. coli* proteins, red dots represent TP yeast proteins, green dots represent TN human proteins, and gray dots represent FP or FN proteins. For **g–i**, the data were processed through the best method combinations without imputation. Benchmarks are performed on protein quantification results by DIA-NN. The data are processed starting with SR75. Differential analysis was performed between the S4 and S2 sample groups. Differential proteins are determined with p-value < 0.05 and $|\log_2 \text{FC}| > \log_2 1.2$.

- Only row-based imputation methods can be considered reliable for DE analysis, i.e. if the other methods are also tested for DE, their potential to produce false positives needs to be discussed.

Response: Thank you for the comment. We appreciate your concern regarding the potential for false positives when using other imputation methods for differential expression analysis. While we surveyed several literatures on missing value imputation (*Nucleic Acids Research*, 2020, 48: e83, DOI:10.1093/nar/gkaa498; *Briefings in Bioinformatics*, 2021, 22: bbaa112, DOI:10.1093/bib/bbaa112; *Proteomics*, 2022, 22: e2200092, DOI:10.1002/pmic.202200092; *Nature Communications*, 2024, 15: 3922, DOI:10.1038/s41467-024-47899-w), we did not find reports that non-row-based imputation methods are unreliable for differential analysis. The imputation methods selected in this study are widely used and reported in the literature, which are valid for the context of the analysis.

We acknowledge that reliability is an important consideration for the selection for imputation methods. It has been reported (also in the above literatures) that imputation should be conducted considering the mechanisms from which the missing values have originated. Therefore, we mainly discussed the imputation methods along with the sources of missing values in DIA single-cell datasets. We believe this is appropriate for the scope of this study.

Thank you again for your insightful comment.

- What was the difference in LC-MS state/settings between the batches? Are any proteins differentially expressed between the batches or the variation between batches is negligible and does not model batch effects in real experiments?

Response: Thank you for the questions. In the revised version, differential proteins between the batches were added to the result table (**Supplementary Data 3** and **4**). For the high-performing method combinations, no differential proteins were found between the batches.

To better reflect the variation in real experiments, we performed another experiment to mimic single-cell proteome samples which are actually prepared from very little amount of proteins subjected to independent digestion. Three batches of data were generated, resulting in a dataset containing two batches of samples preparation and another one containing two batches differing in LC-MS instruments (timsTOF Pro 2 and timsTOF Pro, **Fig. 3a**). We believe this experiment can model batch effects in real experiments better.

Corresponding revision was made on page 13 of the main text.

- Please use PCA instead or in addition to UMAP for visualizing batches, UMAP interpretation is highly problematic.

Response: Thank you for the comment. In the revised version, we have change to PCA for visualizing the batch correction results.

- Plots like on Figure 2f are difficult to interpret, I recommend authors either remove them or visualize differently.

Response: Thank you for the suggestion. We are sorry for the confusing presentation in the previous manuscript. In the revised version, the figures were re-organized. In **Fig. 2b**, the parallel coordinate representation shows metrics using different method combinations, where the number of highlighted method combinations are reduced to top 5% and the top 3 are overstriking. In **Figs. 2c** and **2d**, the metrics are visualized in a hyperbox, the method choice in each step of the best method combination is marked with dashed lines.

In order to show which steps would affect the result to what extent, we further fitted the results using a learning-to-rank model and SHAP explanations (**Supplementary Fig. 17**). The resulting SHAP values quantify the contribution of each feature (method) to a prediction, and the SHAP interaction values extend this by measuring how pairs of features (methods of two steps) jointly influence predictions, capturing their combined effects beyond individual contributions. Corresponding revision was made on page 12 of the main text.

We expect that with these modifications, the data presentation has been more informative and concise.

Supplementary Figure 17. Explanations of the patterns of high-performing method combinations (S4/S2 SR75)

a Feature importance of the model. **b** Mean absolute SHAP values for each step. **c** SHAP values for the method choices in each step. **d** Mean absolute SHAP interaction values for each two steps. **e** SHAP interaction values for pairwise method choices in each two steps. In **c** and **e**, the boxes mark the first and third quartile and the lines inside the boxes mark the median; the whiskers extend from the box to the farthest point lying within 1.5 times the inter-quartile range. Individual data points are overlaid as dots. The median values and frequency (*n*) are indicated for each method choice. The data are processed starting with SR75. Differential analysis was performed between the S4 and S2 sample groups. Differential proteins are determined with p-value < 0.05 and |log₂ FC| > log₂ 1.2. Only the top 25% method combinations are subjected to SHAP explanation and the method choices with *n* < 3 are not shown. Source data are provided as a Source Data file.

- The authors should establish the robustness of the ARI and purity metrics at least by (i) comparing the metric values between the cases when different individual runs are excluded from consideration and, preferably, also by (ii) splitting the set of runs in two groups, randomly but in a balanced fashion, and evaluating the degree of similarity of the metrics calculated separately for each group.

Response: Thank you for the comment. In the revised version, we evaluated the robustness of the ARI and other metrics (purity was no longer used) by cross validation.

(1) Leave-one-out cross validation. For each time n from 1 to 6, the n -th run in each batch and each sample group was excluded, resulting in 6 datasets. The Pearson correlation coefficient (PCC) of the ranks and metrics of method combinations was computed in 10 pairwise comparisons of the results of 6 datasets for each of the 4 sparsity reduction criteria (resulting in 60 values in total). A median PCC of 0.91 was achieved for the ARI values and 0.94 for the ranks (**Supplementary Fig. 13a**).

(2) Split-half cross validation. The runs in each batch and each sample group were randomly split into two balanced groups, forming two subsets. The PCC values were computed between the results of the two subsets for each of the 4 sparsity reduction criteria. This procedure was repeated 10 times (resulting in 40 values in total). A median PCC of 0.78 was achieved for the ARI values and 0.86 for the ranks (**Supplementary Fig. 13b**).

The results indicated the robustness of the metrics and ranking scheme. Corresponding revision was made on page 11 of the main text and **Supplementary Note 7**.

- The authors should discuss whether clustering-based metrics such as ARI and purity are useful in practice. Missing values and imputation are the major factors affecting clustering, however they do not have an effect on the detection of proteins with complete profiles as DE. I believe the number of true/false DE proteins is a significantly more reliable metric and should be used as a basis for imputation, normalisation and batch correction methods comparison instead or in addition to ARI/purity.

Response: Thank you for the comment. We agree that metrics based on the number of true/false differential proteins is more reliable as they reflect the performance of the workflow for differential expression analysis in practice. We included clustering-based metrics such as ARI because they are commonly used to evaluate the performance of batch effect correction in single-cell omics (*Genome Biology*, 2020, 21: 12, DOI: 10.1186/s13059-019-1850-9). In the revised version, the method combinations were compared in a comprehensive consideration of ARI, pAUC, and F1-score. This scheme fits the principle that metrics on true/false differential proteins are used as a basis for comparison in addition to clustering-based metrics.

We explained this in the discussion on page 18 of the main text.

Supplementary Figure 13. Evaluation of the robustness of the metrics and ranking scheme by cross validation **a** Correlations of the ranks and metrics of the method combinations between results obtained by leave-one-out cross validation. **b** Correlations between results obtained by split-half cross validation. The boxes mark the first and third quantile and the lines inside the boxes mark the median; the whiskers extend from the box to the farthest point lying within 1.5 times the inter-quartile range. Individual data points are overlaid as dots. The median values are indicated. Source data are provided as a Source Data file.

Reviewer #3:

In their manuscript “Benchmarking informatics workflows for data-independent acquisition single-cell proteomics” Wang et al. present a comprehensive benchmark of data analysis workflows for single-cell proteomics. Using a simulated sample consisting of mixed proteomes and real single-cell samples with a spike-in scheme, they benchmark different steps during data analysis including DIA data analysis software, sparsity reduction, missing value imputation, normalization, batch effect correction, and differential expression analysis. The manuscript is well structured and presented. I agree with the use of mixed species samples, however I have some concerns regarding the data analysis. The benchmarks using the real single-cell samples are a good idea to include chemical noise to the analysis, however I have concerns about the way the data is analyzed.

Response to general comments: Thank you very much for taking the time and effort to review our manuscript. We really appreciate the valuable comments and suggestions, and have modified our manuscript accordingly.

Major points:

1. Line 49: Different DIA data analysis solutions have indeed been compared previously in <https://doi.org/10.1038/s41467-024-52605-x> and <https://doi.org/10.1016/j.mcpro.2024.100839>

The authors should thus tone down their statements accordingly and cite these papers.

Response: Thank you for the comment. In the revised version, we have cited these papers and changed the statements.

“Different DIA data analysis solutions have been compared in a systematic way for bulk proteomics, and this has been extended to the single-cell level in a few recent studies^{32,33}.”

2. To compare quantitative accuracy of their workflows, the authors use violin plots that describe the distribution of measured fold-changes. Then they compare the deviation of the median of these distributions to the true fold-change to highlight the best method. These comparisons are very close throughout the manuscript and thus, mostly not very conclusive. Unfortunately, the median does not measure the spread of the fold-changes, which is quite important. Furthermore, many of the comparisons between search-engines could mainly be driven by the different number of precursors and the non-overlapping precursors. For example, if more low-abundant precursors are identified, these could decrease the overall accuracy although the overlapping precursors are equally well quantified.

The authors should compare quantitative accuracy using a measure that includes the spread of the fold-changes. E.g. the distributions of the absolute fold-change errors. Furthermore, the authors should make sure that the differences between search engines are not produced by differences in the number of precursors or the non-overlapping precursors by repeating their comparison with only overlapping precursors.

Response: Thank you for the comment. In the revised version, when comparing the quantitative accuracy of different software or searching strategies, only the overlapping analytes were considered. To include the spread of the fold changes, statistical tests were performed on the \log_2 FC distributions. In addition to the medians of \log_2 FC errors, comparison pairs with t-test p-value < 0.05 and Cohen's $|d| > 0.2$ were considered as significant differences. These criteria excluded the comparison that were very close and make the reported results more conclusive. **Figure 1** and the related **Supplementary Figs.** were updated to include the p-values. For the convenience of a reader, critical data of these comparison is summarized in **Supplementary Data 1**. Corresponding revision was made on pages 6 and 7 of the main text.

3. In Line 182, the authors show very high fractions of false transfers to E. coli data from human and yeast. Can the authors rule out the possibility that these peptides are indeed present in the sample due to carry-over on the liquid chromatography column from previous samples containing human and yeast?

Response: Thank you for the question. Considering the identified entrapment analytes that were originated from potential common contaminants, blank samples were inserted among the single-organism samples. For the peptide standard samples, 3 blank LC-MS injections were performed; for the independent-digestion samples, 3 blank samples were subjected to digestion before LC-MS analysis and another 3 blank LC-MS injections were performed. The entrapment analytes identified from the blank samples were fewer than those from the single-organism samples, indicating the latter (at least a large proportion of them) were not among the contaminants and not present in the sample (**Supplementary Figs. 8a and 9a**).

Corresponding revision was made in **Supplementary Note 3**.

Supplementary Figure 8. Comparison of false positive detection by different software using the library free searching strategy at the protein level.

a Numbers of quantified proteins per run. For each sample, correctly detected proteins should be from the organism specific to the sample (in green), while those from other organisms (in red) are potential false positives. Results of blank injections are shown to assess potential contaminants. **b** Numbers of organism-matched (in green) and potential false positive (in red) proteins quantified in at least specified percentages (data completeness) of runs.

4. For the batch correction, did the authors provide biological covariates to the methods? For Combat that would be the model matrix where one can specify the outcome of interest. As far as I know, Scanorama does not use biological covariates, which would explain its lower performance. If the authors provided biological covariates, it would be important to point out this difference and show how Combat and limma perform if no biological covariates are provided to the model. These covariates are not always available in single-cell data.

Response: Thank you for the comment. In the revised version, we clarified that covariates were provided to limma and Combat. In addition, we tested how limma and Combat perform if no biological covariates are provided to the model.

In the original method combinations, limma, Combat-P, and Combat-NP were replaced with their no-covariate versions (denoted as limma-NC, Combat-P-NC, and Combat-NP-NC). To compare the performance with and without covariates, benchmarking results without covariates were ranked combined with the original ones. We focused on the results of S4/S2 with SR75. The method combinations were split into 3 groups, i.e., limma/Combat with covariates, limma/Combat without covariates, and Scanorama and NoBC, and the top 1% method combinations within each group are visualized in **Supplementary Data 3** (panels b and c in Figures SD3-22 for DIA-NN, SD3-23 for Spectronaut, SD3-24 for PEAKS). The no-covariate versions of limma/Combat underperformed those with covariates with a slight gap, while they were still better than Scanorama.

To simulate the scenario that covariates are not available, benchmarking results without covariates (using limma-NC, Combat-P-NC, Combat-NP-NC, Scanorama, and NoBC) were ranked. The variation of the metrics with different method choices are visualized in Figures SD3-22, SD3-23, and SD3-24 (panels d, e, and f). The method combinations with limma-NC yielded high ARI and pAUC values.

Corresponding revision was made on page 12 of the main text and **Supplementary Note 9**.

a Batch effect correction

S4/S2 SR75 p-value < 0.05 $|\log_2 FC| > \log_2 1.2$

b

c

d ARI

e pAUC

f

Best method combination (no covariates): KNN Sum limma-NC DESeq2

g

h

i

(Legend on next page)

Figure SD3-22. Performance comparison of method combinations whether covariates are provided for batch effect correction (DIA-NN S4/S2 SR75)

a The evaluated batch effect correction methods with or without covariate support. **b** Parallel coordinate representation showing metrics using different method combinations with or without covariates, ranked together. **c** Compositions of the top 1% and 5% method combinations in **b**. Mappings of the serial numbers to detailed methods for other steps are present in Fig. 2a. **d** Adjusted Rand index (ARI) metrics. **e** Partial area under receiver operator characteristic curve (pAUC) metrics. In **d** and **e**, the metrics are visualized in a hyperbox, where each face displays the metrics with two steps variable and the other steps fixed to those of the best method combination without covariates. For the best method combination, the method choice in each step is marked with dashed lines. Dot sizes and colors indicate the metric values. **f** Accuracy (dots), recall (triangles), precision (squares), and F1-score (bars) metrics. Rows represent batch effect correction methods and columns represent normalization methods. The other steps are those of the best method combination. **g** Clustering result of the 5 groups of samples visualized using principal component analysis for dimension reduction. The fill colors indicate the sample groups and the shape indicate the batches. The border colors indicate the clusters. **h** Receiver operator characteristic (ROC) curves using $-\log_{10}$ p-value as scores. The optimal cut-offs with false positive rate (FPR) ≤ 0.1 are marked using black dots with score threshold (Thr), FPR, and true positive rate (TPR) values indicated. **i** Volcano plots. Blue dots represent TP *E. coli* proteins, red dots represent TP yeast proteins, green dots represent TN human proteins, and gray dots represent FP or FN proteins. For **f-h**, the data were processed through the best method combinations without covariates. Benchmarks are performed on protein quantification results by DIA-NN. The data are processed starting with SR75. Differential analysis was performed between the S4 and S2 sample groups. Differential proteins are determined with p-value < 0.05 and $|\log_2 \text{FC}| > \log_2 1.2$.

5. Regarding the differential expression benchmarks. A common strategy in proteomics is to ignore missing values for DE testing. Did the authors consider including that into the comparisons?

Response: Thank you for the comment. In the revised version, we tested the performance with no imputation whenever allowed by the batch correction and statistical test methods used. In these benchmarking workflows, missing values were kept as they were (KeepNA). Batch correction methods included NoBC and limma, and statistical test methods included t-test, Wilcox, and limma-trend. Other methods were excluded since they do not support missing values.

To compare the performance with and without imputation, benchmarking results with KeepNA were ranked combined with the original ones. We focused on the results of S4/S2 with SR75. The top 1% method combinations with and without imputation are visualized in **Supplementary Data 3** (Figures SD3-46 for DIA-NN, SD3-47 for Spectronaut, SD3-48 for PEAKS), showing the gap between them. The low performance of the method combinations without imputation may be due to the limited choices of batch correction and statistical test methods. Other the other hand, with the method choices in other steps fixed, the benchmarking metrics without imputation were still lower than those with imputation.

Corresponding revision was made on page 12 of the main text and **Supplementary Note 10**.

6. Lines 597 & 639: Did the authors use the same number of clusters across comparisons to calculate ARI? How stable are these results across different numbers of clusters?

Response: Thank you for the comment. For performance evaluation of batch effect correction, the samples were clustered using the Louvain algorithm implemented in the R package Seurat. The clustering function FindClusters has a resolution parameter to control the number of communities (larger using a value above 1.0 or smaller using a value below 1.0).

In the revised version, we surveyed the resolution from 0.6 to 1.4 stepped by 0.2. The Pearson correlation coefficient (PCC) of the ranks and metrics of method combinations was computed across the resolution parameter. A median PCC of 0.96 was achieved for the ARI values and 0.98 for the ranks (**Supplementary Fig. 10**), indicating that ranking was stable regardless of the resolution parameter. Therefore, a resolution parameter of 1.0 was used in all the analyses.

Corresponding revision was made on page 10 of the main text and **Supplementary Note 5**.

(Legend on next page)

Figure SD3-46. Performance comparison of method combinations with and without imputation (DIA-NN S4/S2 SR75)

a The evaluated method combinations. Method choices in blue can process data without imputation (containing missing values) whereas those in gray cannot. **b** Parallel coordinate representation showing metrics using different method combinations with or without imputation, ranked together. **c** Compositions of the top 1% and 5% method combinations in **b**. Mappings of the serial numbers to detailed methods for each step are present in **a**. **d** Adjusted Rand index (ARI) metrics. **e** Partial area under receiver operator characteristic curve (pAUC) metrics. In **d** and **e**, the metrics are visualized in a hyperbox, where each face displays the metrics with two steps variable and the other steps fixed to those of the best method combination without imputation. For the best method combination, the method choice in each step is marked with dashed lines. Dot sizes and colors indicate the metric values. **f** Accuracy (dots), recall (triangles), precision (squares), and F1-score (bars) metrics. Rows represent batch effect correction methods and columns represent normalization methods. The other steps are those of the best method combination. **g** Clustering result of the 5 groups of samples visualized using principal component analysis for dimension reduction. The fill colors indicate the sample groups and the shape indicate the batches. The border colors indicate the clusters. **h** Receiver operator characteristic (ROC) curves using $-\log_{10}$ p-value as scores. The optimal cut-offs with false positive rate (FPR) ≤ 0.1 are marked using black dots with score threshold (Thr), FPR, and true positive rate (TPR) values indicated. **i** Volcano plots. Blue dots represent TP *E. coli* proteins, red dots represent TP yeast proteins, green dots represent TN human proteins, and gray dots represent FP or FN proteins. For **g–i**, the data were processed through the best method combinations without imputation. Benchmarks are performed on protein quantification results by DIA-NN. The data are processed starting with SR75. Differential analysis was performed between the S4 and S2 sample groups. Differential proteins are determined with p-value < 0.05 and $|\log_2 \text{FC}| > \log_2 1.2$.

Supplementary Figure 10. Comparison of results with different clustering parameters

Correlations of the ranks and metrics of the method combinations between results obtained under varying resolution parameters (0.6, 0.8, 1.0, 1.2, 1.4), holding constant both the comparison group and the sparsity reduction conditions. The boxes mark the first and third quartile and the lines inside the boxes mark the median; the whiskers extend from the box to the farthest point lying within 1.5 times the inter-quartile range. Individual data points are overlaid as dots. The median values are indicated. Source data are provided as a Source Data file.

7. For the real single-cell dataset, the most relevant normalization method should be no-normalization, because the ratio spike-in is constant and only the single-cell background is variable. All the applied normalization methods are not compatible with these samples, because they try to align distributions that are different by experimental design. The ratios will be distorted by the size of the single cell; however, the size of the single-cell is not relevant for this experiment. Maybe the batch-correction step corrects for many of these un-suitable normalizations that were applied.

Apart from no-normalization, one could normalize only based on the spike-in peptides, which are assumed to have the same total amounts across samples. Thus, sum-normalization should perform well here. Going even further, all processing steps should only be applied on a matrix containing only the spike-in peptides, because also the batch-correction will work with the assumption that the real distributions of the samples are comparable, which they are not due to the variable single-cell sizes. The single-cells should only provide the chemical background that challenges the overall analysis.

Response: Thank you for the comment. We agree that the single-cells should only provide the chemical background and the spike-in peptides are the main part for benchmarking. However, applied on a matrix containing only the spike-in peptides will not be suitable for benchmarking because all the spike-in proteins should be differential proteins and there are no negatives to calculate the metrics. To avoid the distortion of ratios by the size of the single cells, in the revised version, we selected a subset of human proteins, including histones with cell size-independent copy numbers (*Molecular & Cellular Proteomics*, 2014, 13: 3497–506, DOI:10.1074/mcp.M113.037309; *Journal Proteome Research*, 2023, 22: 3773–3779, DOI:10.1021/acs.jproteome.3c00441), as well as housekeeping proteins with probably stable expression (*Trends in Genetics*, 2013, 29: 569–574, DOI:10.1016/j.tig.2013.05.010; *GigaScience*, 2019, 8: giz106, DOI:10.1093/gigascience/giz106) for the control groups. All the processing steps were applied on a matrix containing only the spike-ins and the subset of human proteins. In this setting, other proteins from the single cells should only provide the chemical background that challenges the overall analysis. The normalization and batch correction will work with the assumption that the real distributions of the samples were comparable, which they were not due to the variable cell sizes.

Corresponding revision was made on pages 14 and 15 of the main text.

Minor points:

1. Lines 23, 82 & 414: this should say “simulated” instead of “stimulated”.

Response: Thank you for the comment. We have corrected these typos.

2. Fig. 1b: From the figure or its caption, it’s not clear that this is from the library-free data.

Response: Thank you for the comment. We have clarified this in the legend.

3. Lines 131-133 – the authors claim that PEAKS quantified more analytes than Spectronaut, but then state that PEAKS quantified 11,165 peptides vs 12,537 peptides in Spectronaut. With peptides being the analyte measured in LC-MS based proteomics, I would argue it is the opposite?

Response: Thank you for the comment. We have reanalyzed the data and updated the results in the revised version.

4. Line 202: How did the authors create these different batches?

Response: Thank you for the question. The original dataset of mixed-organism peptide standards was created with batches of LC-MS analyses on different dates.

To better reflect the variation in real experiments, we performed another experiment to mimic single-cell proteome samples which are actually prepared from very little amount of proteins subjected to independent digestion. Three batches of data were generated, resulting in a dataset containing two batches of samples preparation and another one containing two batches differing in LC-MS instruments (timsTOF Pro 2 and timsTOF Pro, **Fig. 3a**). We believe this experiment can model batch effects in real experiments better.

Corresponding revision was made on page 13 of the main text.

5. Line 433: Working with 75% data completeness is possible for the authors only because they have very similar samples at hand (ie. homogeneous cell lines). I think it would be important to note that this will be significantly more complicated when analyzing many different cell types, where missing values will be much higher due to cell-type specific effects.

Response: Thank you for the comment. In the revised version, we have stated this point in the discussion (on page 19).

“For the homogeneous cell lines in this study, working with 75% data completeness is a good trade-off between gaining detected proteins and reducing the burden of missing value imputation. This will be more complicated when analyzing many different cell types, where missing values will be much higher due to cell-type specific effects.”

6. Line 573: Did the authors deactivate the internal normalization of the search engines or did they use the standard settings? This is important for RT-dependent normalizations, as in DIA-NN this cannot be reversed.

Response: Thank you for the question. We used the standard settings. We did not deactivate the internal normalization for the following considerations.

The study includes two parts. For the first part, internal normalization is crucial for the comparison of quantification precision and accuracy. During the analyses, all of the software workflows were run using the default settings recommended for diaPASEF data with minimized modifications to make their results comparable. For the second parts, the benchmarking workflow includes normalization again because it is a crucial step in the downstream informatic analyses. In a real scenario, users can perform DIA data analysis with the recommended settings by the software (including normalization), and also perform normalization in their informatic workflows (e.g., normalization in Perseus or Seraut). Thus, we regard them as separate process.

We have clarified the use of internal normalization in the revised manuscript.

Responses to the Reviewer's Comments

Benchmarking informatics workflows for data-independent acquisition
single-cell proteomics

Jianwei Wang et al.

Reviewer #1:

In the revision, the authors have made great efforts to improve the manuscript. Only a few minor points remain to be addressed and the reviewer would recommend publication of this pretty comprehensive work after the very minor revision.

Response: Thank you very much for taking the time and effort to review our manuscript. We really appreciate the valuable comments and suggestions, and have modified our manuscript accordingly.

Line 147: Spectronaut should be DIA-NN.

Response: Thank you for the comment. We have corrected this mistake.

In Figure 4C annotation, does the blue bar represent TP and the green is for TN?

Response: Thank you for the comment. As indicated in the figure and the legend, TN the blue bars represent TN and the green bars represent TP.

Line 454: For this key statement “When a project specific spectral library is not available (e.g., the samples are rare or resources for additional injections are not sufficient), the built-in prediction by DIA-NN is preferred for library-free analysis.” it seems contradictory to what are shown in Results. Spectronaut in directDIA yields more protein IDs, higher data completeness and comparable FPRs in differential protein analysis (Figs 1B, 1C, and 4C). So would this point be mentioned and recommended in Discussion?

Response: Thank you for the comment. We have updated the discussion accordingly.

“ Spectronaut directDIA is recommended as it yields more protein identifications, higher data completeness and comparable FPRs in differential protein analysis. The built-in prediction by DIA-NN is also an appropriate choice with high quantification accuracy and precision for library-free analysis.

Reviewer #2:

The authors have revised and significantly improved the manuscript, addressing the most important comments. As noted before, this is an excellent study that should be published, provided several easy to deal with issues are addressed.

Response: Thank you very much for taking the time and effort to review our manuscript. We really appreciate the valuable comments and suggestions, and have modified our manuscript accordingly.

The authors now clarify the software settings that they have applied in their analysis. However, there are two apparent issues. While the authors state protein FDR was controlled at 1%, the pg matrix output of DIA-NN that does not provide protein FDR control was used. How did authors achieve protein FDR control? For peptide-level quantification, the authors averaged precursor quantities. This is unlikely to produce quality results, especially in the context of high missing value rates inherent to SCP. Can the authors use MaxLFQ or Top N method instead?

Response: Thank you for the comment. The pg_matrix report output by DIA-NN has been filtered at 1% FDR, using global q-values for protein groups and both global and run-specific q-values for precursors and an additional 5% run-specific protein-level FDR filter. (<https://github.com/vdemichev/DiaNN?tab=readme-ov-file#output>)

In the revision, we used the main report output by DIA-NN, and applied the following filters: Q.Value (run-specific, at the precursor level) at 0.01 and Lib.Q.Value (for the respective library entry, at the precursor level) at 0.01, as well as PG.Q.Value at 0.05 and Lib.PG.Q.Value at 0.01 (same as above, at the protein group level). (<https://github.com/vdemichev/DiaNN?tab=readme-ov-file#main-output-reference>)

The protein level results are the same as those using the pg_matrix report in our previous submission. For the peptide results, the precursor-level results were aggregated by “Stripped.Sequence” and the quantity of each peptide was the sum of those of the top 3 corresponding precursors. The results are very close to those using the pr_matrix report in our previous submission. Corresponding figures and tables have been updated accordingly.

I noted that human identifications should not be counted as false in FDR tests. The authors argue that removing peptides found in blanks from consideration makes this acceptable. This is incorrect for a number of reasons, one being that the blanks cannot be reliably assumed to represent the degree of contaminant presence in SCP samples. Given that this analysis is not just uninterpretable for the aforementioned reason but also completely unnecessary here, as using non-human entrapment identifications is sufficient, it should be removed.

Response: Thank you for the comment. We have removed the error rate estimation of

using human entrapment identifications. Corresponding figures and tables have been updated accordingly.

Reviewer #3:

I have received this manuscript for review only after the first round of revisions, and had a thorough look at the rebuttal and updated manuscript. I can confirm that my initial concerns were shared by the other reviewers, and have now been appropriately addressed by the authors. I think the manuscript serves as a useful resource for the field, and especially appreciate the use of real single-cell samples as this is critical for establishing 'best practises' in the field.

Therefore, I deem the manuscript suitable for publication in its current state.

Response: Thank you very much for taking the time and effort to review our manuscript. We really appreciate the valuable comments.